# EXPLORING SPARSITY FOR PARAMETER EFFICIENT FINE TUNING USING WAVELETS FOR VISION

## ABSTRACT

Efficiently adapting large pretrained models is critical under tight compute and memory budgets. While PEFT methods like LoRA achieve efficiency through low-rank updates, their discrete rank constraint limits fine-grained parameter control and confines adaptations to low-dimensional subspaces. We propose Wavelet Fine-Tuning (WaveFT), which learns sparse updates in the wavelet domain of weight matrices, enabling fine-grained control over trainable parameters well below LoRA's minimum rank. Wavelet bases provide semi-local receptive fields that aggregate spatially coherent gradients, offering better coverage than direct weight sparsity (SHiRA) without the destructive interference of global Fourier bases (FourierFT). This structure naturally matches vision tasks where gradients are sparse during fine-tuning, since most pretrained weights require minimal adjustment. We provide theoretical analysis showing: (i) sparse methods achieve high-rank updates, avoiding LoRA's subspace bottleneck and enabling higher representational capacity, and (ii) a gradient coverage framework explaining when wavelet-domain adaptation outperforms alternatives. We perform experiments across text-to-image generation (SDXL), image classification (ViT), and language understanding (GLUE). WaveFT demonstrates state-of-the-art results among PEFT methods for vision tasks, where wavelets effectively capture sparse gradient structure through improved coverage, while performing comparably on NLP benchmarks.

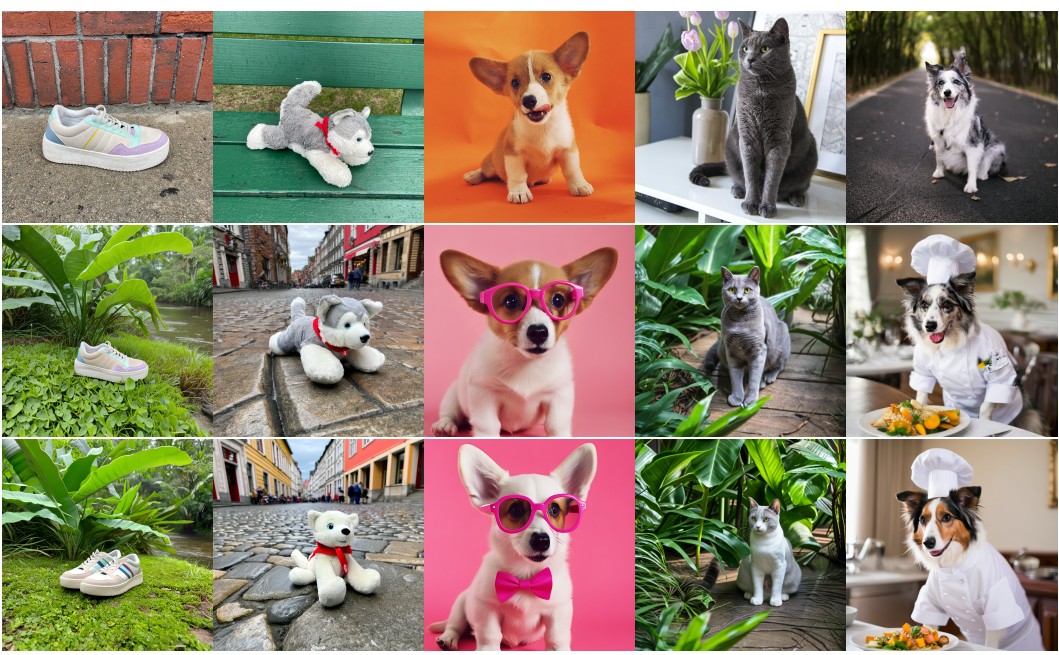

Figure 1: The original images (top row), WaveFT results (middle row), LoRA results (Hu et al., 2022) (bottom row).

# 1 INTRODUCTION

Large-scale diffusion models, *e.g.* Rombach et al. (2022); Podell et al. (2024), represent the state-of-the-art in text-to-image generation and are increasingly deployed across industry applications. Adapting these powerful pre-trained models to specific downstream needs, such as personalizing them for particular subjects or styles, is crucial for maximizing their utility. However, fully fine-tuning these massive models is often computationally infeasible due to significant memory requirements, compute costs, and storage needs. Parameter-Efficient Fine-Tuning (PEFT) techniques offer a compelling solution by adapting models through training only a small subset of parameters.

Among PEFT methods, Low-Rank Adaptation (LoRA) (Hu et al., 2022) has gained widespread popularity, demonstrating strong performance by learning low-rank updates to weight matrices. Despite its success, LoRA's reliance on an integer rank $r \geq 1$ imposes two limitations: the minimum rank forces allocating more parameters than necessary for simple adaptations, and discrete rank increments prevent fine-grained control across layers.

We introduce Wavelet Fine-Tuning (WaveFT), which learns a sparse set of $p$ parameters in the *wavelet domain* representation of the weight update matrix $\Delta W$. These learned coefficients are transformed back to the weight domain via the Inverse Discrete Wavelet Transform (IDWT). As a baseline to isolate the effect of the wavelet transform, we compare against SHiRA (Bhardwaj et al., 2024), which applies sparse updates directly in the weight domain. The sparse parameterization governed by $p$ permits fine-grained adjustment of the adaptation budget, enabling parameter counts well below LoRA's minimum at $r = 1$.

A natural question arises: *why should the wavelet domain be preferable to direct weight-space sparsity?* We provide two complementary answers. First, both WaveFT and SHiRA produce high-rank updates, escaping LoRA's "subspace bottleneck" that confines modifications to an $r$-dimensional subspace (Section 3); we demonstrate that this higher representational capacity translates to more diverse outputs in image generation. Second, and more critically for differentiating WaveFT from SHiRA, wavelet coefficients have *semi-local receptive fields* that aggregate gradient information from spatially coherent neighborhoods. Under the sparse gradient conditions typical of fine-tuning, where most pretrained weights are already near-optimal, this yields better gradient coverage than weight-domain sparsity, without the destructive interference that plagues global Fourier bases (Section 4.1).

Our experiments span text-to-image generation (SDXL), image classification (ViT-Base), and language understanding (GLUE). On the DreamBooth benchmark, WaveFT achieves 0.495 DINO similarity versus LoRA's 0.463 at equivalent parameter counts, while also improving output diversity (LPIPS: 0.348 vs 0.309). For image classification, WaveFT attains 78.29% average accuracy with only 72K parameters, outperforming LoRA (77.58% at 581K parameters). On NLP benchmarks, WaveFT performs slightly below global methods like FourierFT, consistent with our theoretical prediction that wavelet advantages emerge primarily under the sparse gradient conditions characteristic of vision tasks.

In summary, our main contributions are:

(i) WaveFT, a PEFT method that learns sparse updates in the wavelet domain, enabling parameter budgets below LoRA's minimum rank while achieving high-rank weight updates;

(ii) A theoretical framework comprising: (a) rank analysis proving sparse methods escape LoRA's subspace bottleneck, yielding higher representational capacity and more diverse outputs, and (b) a gradient coverage framework explaining when wavelet-domain adaptation outperforms alternatives;

(iii) Comprehensive experiments across text-to-image generation (SDXL), image classification (ViT-Base), and language understanding (GLUE), demonstrating state-of-the-art results among PEFT methods for vision tasks;

(iv) Extensive ablation studies revealing: robustness to input permutation (validating that WaveFT's advantage stems from gradient coverage, not spatial structure), superior stability across random seeds compared to SHiRA, consistent performance across wavelet families, and a controllable fidelity-alignment trade-off via the scaling factor $\lambda$.

The remainder of this paper is organized as follows: Section 2 reviews related work on PEFT methods. Section 3 presents the WaveFT method. Section 4 provides theoretical analysis including the gradient coverage framework. Section 5 presents experimental results, and Section 6 concludes.

## 2 RELATED WORK

**PEFT and intrinsic dimensionality.** The feasibility of PEFT is partly motivated by the concept of *intrinsic dimensionality*, suggesting that the essential changes required for downstream tasks might reside in a low-dimensional subspace (Li et al., 2018; Aghajanyan et al., 2021). Aghajanyan et al. (2021) specifically show that fine-tuning large language models (LLMs) effectively occurs within low-dimensional subspaces. While some methods explicitly combine low-rank and sparse updates (Nikdan et al., 2024; Huang et al., 2025; Zhang et al., 2025b), others directly fine-tune only specific components, such as biases or partial connections (Woo et al., 2025).

**LoRA extensions.** LoRA (Hu et al., 2022) is perhaps the most prominent PEFT method, achieving efficiency by representing the weight update $\Delta W$ as a product of two low-rank matrices, $\Delta W = BA$. This low-rank constraint significantly reduces trainable parameters, controlled by the rank $r$. Numerous extensions are proposed to improve LoRA. Some focus on dynamically allocating the parameter budget (rank) based on layer importance (Zhang et al., 2023; Jiang et al., 2025; Zhou et al., 2025) rather than using a fixed rank. Others explore alternative matrix factorizations involving Hadamard or Kronecker products (Hyeon-Woo et al., 2022; YEH et al., 2024; Chavan et al., 2023; Edalati et al., 2022). Significant effort also goes into improving parameter efficiency further through shared parameter schemes (Kopiczko et al., 2024; Li et al., 2024; Jiang et al., 2024; Ding et al., 2025), multi-scale structures (Zhao et al., 2025), summation compression (Quercia et al., 2025), or optimizing shared and specific modules (Nguyen et al., 2025; Zhang et al., 2025c). Other works delve into the internal mechanics, analyzing the asymmetry of LoRA matrices (Zhu et al., 2024), decomposing weights differently (Liu et al., 2024a), optimizing training dynamics (Hayou et al., 2024; Lialin et al., 2024; Shi et al., 2024), or using weight guidance (Kang, 2024). While these methods enhance LoRA, they typically retain the core low-rank decomposition and the limitation of discrete rank control. Our approach fundamentally differs by using direct sparsity parameterization ($p$) instead of rank ($r$), allowing finer budget control.

**Transformed parameterizations.** Several recent methods explore adapting models by operating in domains other than the standard weight space. FourierFT (Gao et al., 2024) learns sparse updates in the 2D discrete Fourier domain, while FouRA (Borse et al., 2024) applies 1D Fourier transforms to embeddings before LoRA. The proposed WaveFT shares the spirit of operating in a transformed domain but specifically utilizes the wavelet domain. Other related directions include methods that directly adapt components derived from Singular Value Decomposition (SVD) of weights, such as singular values or vectors (Zhang & Pilanci, 2024; Elsayed et al., 2025; Hegde et al., 2025; Bałazy et al., 2024), use deconvolution in subspaces (Zhang et al., 2025a), or constraining the fine-tuning updates to be orthogonal transformations (Qiu et al., 2023; Liu et al., 2024b; Ma et al., 2024).

**Sparse Fine-Tuning.** Sparse fine-tuning methods update only a small, fixed subset of model weights to achieve parameter-efficient adaptation. Earlier work includes DiffPruning (Guo et al., 2021), Fisher Mask (Sung et al., 2021), LT-SFT and Composable SFT (Ansell et al., 2023), which use various masking strategies to select individual weights. Recent approaches like SpIEL iteratively grow and prune indices (Ansell et al., 2024), SMT partitions weights into blocks for gradient-based selection (He et al., 2025), SHiRA explores various types of sparse masks and demonstrates: (a) significantly better performance than LoRA, and (b) reduced concept loss in multi-adapter usecases (Bhardwaj et al., 2024), and SaRA identifies low-magnitude weights for progressive sparse adaptation (Hu et al., 2025).

As direct relevant baselines, we include *SHiRA-rand* in (Bhardwaj et al., 2024) (we refer to as SHiRA for brevity) learning random sparse updates in the weight domain, and FourierFT (Gao et al., 2024), which selects trainable parameters uniformly at random in the fourier domain.

Finally, we note that while many PEFT methods demonstrate success primarily on NLP tasks, their effectiveness and characteristics can differ in the vision domain (YEH et al., 2024). Our work provides a thorough evaluation of sparse adaptation methods (SHiRA, WaveFT) across text-to-image personalization, image classification, and language understanding, with comparisons against strong low-rank and structured PEFT baselines. This cross-domain evaluation enables us to validate our theoretical predictions about when wavelet-domain adaptation excels.

## 3 SPARSE FINE-TUNING IN THE WAVELET DOMAIN

Large pre-trained models are adapted by adding a small update matrix $\Delta W$ to the original parameters $W_0 \in \mathbb{R}^{m \times n}$ with weight $\lambda$, such that

$$W = W_0 + \lambda \Delta W.$$

In the case of LoRA, $\Delta W_{\text{LoRA}}$ is constrained to be a low-rank matrix. Our proposed method instead focuses on learning $\Delta W$ with a sparse parameterization in a way that allows controlling the number of trainable parameters in a fine-grained manner.

More specifically, WaveFT learns a sparse set of parameters within the *wavelet domain* representation of the weight update matrix. The update $\Delta W_{\text{WaveFT}}$ is obtained by applying the 2-Dimensional Inverse Discrete Wavelet Transform (IDWT) to a sparse coefficient matrix $C \in \mathbb{R}^{m \times n}$:

$$\Delta W_{\text{WaveFT}} = \text{IDWT}(C)$$

The matrix $C$ contains the trainable parameters in the wavelet domain. It is constructed to be sparse: $C$ is initialized as a zero matrix ($\mathbf{0}$), ensuring $W = W_0$ at the start of training. We investigate alternative initializations, such as sampling the $p$ trainable parameters from a Gaussian distribution, in our experiments (Section 5).

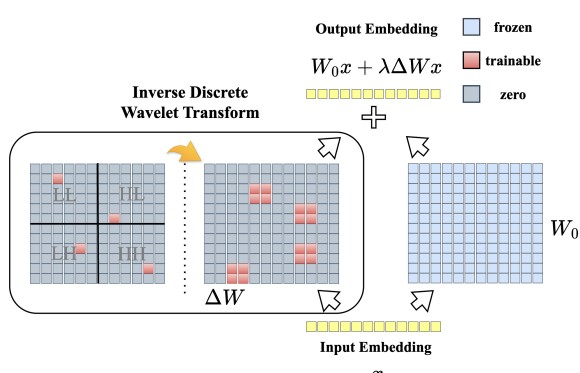

Figure 2: Overview of the proposed method.

We then select exactly $p$ entries of $C$ uniformly at random to serve as trainable parameters. The number of trainable parameters per layer, $p$, is chosen for parameter efficiency (e.g., to match LoRA $r = 1$ or be even smaller). In our standard setup, the budget $p$ is fixed for all adapted layers, though adaptive allocation is explored in Section 5. During optimization, gradients are computed only for the $p$ trainable entries in $C$.

We hypothesize that the structure introduced by the wavelet transform provides an effective parameterization for highly sparse $p \ll m \cdot n$ choices. In our experiments, we thoroughly investigate the validity of this hypothesis with comparisons to unstructured weight-space sparsity, *i.e.* SHiRA (Bhardwaj et al., 2024), or global low-rank approximations, *e.g.* LoRA (Hu et al., 2022). We note that SHiRA can be expressed as a special case of WaveFT via replacing IDWT operators in WaveFT by identity mapping.

This approach of selecting $p$ specific entries for training, as in SHiRA and our WaveFT, can be viewed through the lens of intrinsic dimensionality (Aghajanyan et al., 2021). By fixing all but $p$ randomly chosen elements of $\Delta W$ (for SHiRA) or $C$ (for WaveFT) to zero, we effectively restrict the optimization to a $p$-dimensional subspace of the full $m \times n$ dimensional space, spanned by the standard basis vectors corresponding to these $p$ chosen entries. For WaveFT, a subsequent linear transformation (the IDWT) is then applied to the parameters residing in this sparsely defined subspace.

**Inference efficiency.** WaveFT provides efficient inference. After training, the learned update $\Delta W$ (either $\Delta W_{\text{WaveFT}} = \text{IDWT}(C)$ or $\Delta W_{\text{SHiRA}} = C$) can be computed once and merged with the original weights:

$$W_{\text{final}} = W_0 + \lambda \Delta W$$

Using $W_{\text{final}}$ incurs no inference latency overhead compared to the original model $W_0$. This ensures that the inference speed after merging the adapter is identical to that of the original pre-trained model.

## 4 THEORETICAL ANALYSIS

In this section, we present a series of lemmas that collectively build an argument for the enhanced representational capacity of WaveFT and SHiRA, which stems from their ability to realize high-rank updates to the model weights. This increased capacity is hypothesized to be the foundation for the

observed diversity. We then proceed to give insights on the differences between learning sparse updates in different transformed domains. This characterization helps us understand the performance differences across tasks and domains. The empirical results (presented in Section 5) support these theoretical insights. Most noticeably, Table 2 demonstrate that high-rank methods, WaveFT and SHiRA (Bhardwaj et al., 2024), produce more diverse outputs when generating new images compared to other parameter-efficient fine-tuning techniques, including LoRA (Hu et al., 2022). We also provide insights on why the semi-local structure of wavelets are expected to outperform fully global methods, such as FourierFT (Gao et al., 2024), and fully local methods (SHiRA), as also empirically observed in Section 4.1 on most vision tasks, and why FourierFT may be more suitable in the NLP tasks.

At the core of our analysis is the use of a sparse update matrix. Lemma 1 (Frieze & Pittel, 2004; Erdős & Rényi, 1964) offers fundamental insight into the rank properties of such matrices when constructed with randomly selected non-zero entries.

**Lemma 1.** *Let $A_n$ be an $n \times n$ matrix whose entries are initially all zero. Suppose $p = n(\ln n + c_n)$ distinct positions are chosen uniformly at random from the $n^2$ available positions in $A_n$. These $p$ chosen positions are then filled with random non-zero values.*

*The probability that the resulting matrix $A_n$ is full rank satisfies:*

$$\lim_{n \to \infty} \mathbf{Pr}(A_n \text{ is full rank}) = \begin{cases} 0 & \text{if } c_n \to -\infty, \\ e^{-2e^{-c}} & \text{if } c_n \to c, \\ 1 & \text{if } c_n \to \infty. \end{cases}$$

Lemma 1 provides an asymptotic guarantee: a sufficiently sparse random matrix $\Delta W$ (where *sufficiently sparse* implies the number of non-zero elements $p$ is at least $n(\ln n + c_n)$) is highly likely to be full rank as matrix dimensions grow. Given that the probability of an $m \times n$ matrix with $p$ nonzero entries being full rank is higher than $n \times n$ matrix (for $m \geq n$), this Lemma 1 also holds for $m \times n$ matrices. In SHiRA (Bhardwaj et al., 2024), the update $\Delta W_{\text{SHiRA}}$ is precisely such a sparse matrix, where $p$ entries are randomly chosen to be trainable.

Figure 3 empirically shows the rank of a randomly generated sparse matrix (denoted $\hat{r}$) versus the number of non-zero elements $p$ for attention layer matrix dimensions in the SDXL model (Podell et al., 2024). 95% confidence intervals are shown as shaded. Vertical dashed lines indicate LoRA complexity levels for varying $r$ values according to the corresponding number trainable parameters. The figure demonstrates that the resulting matrix rank rapidly increases as a function of $p$ as asymptotically predicted by Lemma 1 and reaches full rank at a parameter complexity that would correspond to a LoRA adapter with $r = 3$. Thus, we can operate with high confidence that $\Delta W_{\text{SHiRA}}$ is high-rank.

The WaveFT method, which applies an update $\Delta W_{\text{WaveFT}} = \text{IDWT}(C)$, also begins with a sparse matrix $C$ in the wavelet domain. This matrix $C$ is constructed identically to $\Delta W_{\text{SHiRA}}$: $p$ randomly selected coefficients are made trainable, while the rest are zero. The Inverse Discrete Wavelet Transform (IDWT) is a linear transformation that preserves rank. Consequently, the high-rank property established for $C$ (supported by Lemma 1 and Figure 3) directly implies that $\Delta W_{\text{WaveFT}}$ will also be high-rank. Thus, from a rank perspective, WaveFT and SHiRA have equivalent representational capacity; the distinction between them arises from gradient dynamics during training, which we analyze in Section 4.1.

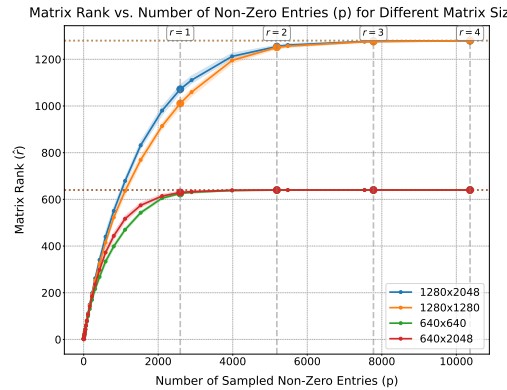

Figure 3: Rank analysis random sparse matrices.

This inherent characteristic of SHiRA and WaveFT, their tendency to produce high-rank updates, contrasts starkly with methods like LoRA, which are explicitly designed to yield low-rank updates. Lemma 2 formalizes this critical difference and its implications. [1]

---

[1]Proofs are in Section A.1

**Lemma 2** (Subspace Bottleneck of LoRA). *For a rank-$r$ adapter update matrix of the form $\Delta W = BA^\top$, where $B \in \mathbb{R}^{m \times r}$ and $A \in \mathbb{R}^{n \times r}$, the following properties hold:*

    *1. The image (column space) of $\Delta W$ is contained within the span of the columns of $B$:*

$$\text{im}(\Delta W) = \{\Delta W x \mid x \in \mathbb{R}^n\} \subseteq \text{span}(\textit{columns of } B).$$

    *2. The kernel (null space) of $\Delta W$ contains the orthogonal complement of the span of the columns of $A$:*

$$\ker(\Delta W) = \{x \in \mathbb{R}^n \mid \Delta W x = 0\} \supseteq (\text{span}(\textit{columns of } A))^\perp = \ker(A^\top).$$

*Consequently, any update $\Delta W$ achieved through such a factorization can only modify the network's activations within the $r$-dimensional subspace spanned by the columns of $B$. Directions orthogonal to the columns of $A$ in the input space are mapped to zero.*

Lemma 2 clearly illustrates that LoRA constrains the update $\Delta W$ to a low-rank structure. As a result, any changes LoRA makes to the model's behavior are confined to a low-dimensional subspace, specifically, the $r$-dimensional space spanned by the columns of its $B$ matrix. This *subspace bottleneck* inherently limits the range and complexity of modifications LoRA can represent.

Having established that our methods produce high-rank updates while LoRA is confined to low-rank ones, we now consider the representational power that high-rank sparse matrices offer:

**Lemma 3** (Block-Sparse Interpolation Capacity). *Let $W_0 \in \mathbb{R}^{m \times n}$ be any fixed matrix. Let*

$$\{x^{(1)}, \ldots, x^{(k)}\} \subset \mathbb{R}^n$$

*be linearly independent, and let arbitrary targets $\{y^{(1)}, \ldots, y^{(k)}\} \subset \mathbb{R}^m$ be given. Set*

$$X = \begin{bmatrix} x^{(1)} & \cdots & x^{(k)} \end{bmatrix} \in \mathbb{R}^{n \times k}, \quad Z = \begin{bmatrix} y^{(1)} - W_0 x^{(1)} & \cdots & y^{(k)} - W_0 x^{(k)} \end{bmatrix} \in \mathbb{R}^{m \times k}.$$

*Let $S \subset [m] \times [n]$ be a fixed sparse support pattern, and define*

$$R = \{\, i \in [m] \mid Z_{i,:} \neq 0\}, \quad S_i = \{\, j \in [n] \mid (i,j) \in S\}.$$

*Assume:*

    *1. $\text{rank}(X) = k$.*

    *2. There exists a single index set*

$$C = \{c_1, \ldots, c_k\} \subset [n]$$

    *such that $X_{C,:} \in \mathbb{R}^{k \times k}$ is invertible and $C \subset S_i$ for every $i \in R$.*

*Then one can construct $\Delta W \in \mathbb{R}^{m \times n}$ with*

    *1. $\text{supp}(\Delta W) \subseteq S$.*

    *2. $(W_0 + \Delta W)\, x^{(l)} = y^{(l)}$ for all $l = 1, \ldots, k$.*

    *3. $\text{rank}(\Delta W) = \text{rank}(Z_R)$, where $Z_R$ is the submatrix of $Z$ restricted to rows in $R$.*

Lemma 3 provides a powerful insight: if a sparse support pattern $S$ is suitably structured relative to a set of $k$ linearly independent inputs $x^{(l)}$ and desired outputs $y^{(l)}$, then a $\Delta W$ confined to this support $S$ can perfectly interpolate these target transformations. Specifically, Hypothesis 2 of the lemma requires that for all rows $i$ where a change is needed (i.e., $i \in R$), the sparse support $S_i$ in that row must contain a common set of $k$ column indices $C$ such that the input submatrix $X_{C,:}$ is invertible. When these conditions hold, an update $\Delta W$ can be constructed that not only matches the desired input–output behavior but whose rank is at least that of the necessary change $Z_R$.

While Lemma 3 considers a fixed support $S$, our methods utilize randomly generated sparse supports. The connection arises because the high probability of achieving a high-rank update matrix (as established by Lemma 1 and Figure 3, and empirically validated in Appendix A.3.1) implies that

the randomly chosen support $S$ is rich enough to allow for the construction of such a $\Delta W$ for a substantial number of target transformations. If the desired set of transformations $\{(x^{(l)}, y^{(l)})\}_{l=1}^{k}$ requires a high-rank $Z_R$ (representing diverse and complex changes), then the resulting $\Delta W$ must also be high-rank. Our methods inherently produce such high rank updates, suggesting they possess greater capacity to represent complex, high-dimensional changes.

This theoretical framework explains the increased output diversity observed with our methods. The ability to operate effectively in a much higher-dimensional modification space means SHiRA and WaveFT are not confined by the "subspace bottleneck" of LoRA. They can represent a richer, more varied family of transformations from the base model. When applied to image generation, this expanded representational power allows the fine-tuned model to explore a broader manifold of possible outputs. Rather than being restricted to changes along only $r$ fixed "directions" as in LoRA, our methods can combine learned sparse parameters to produce a wider array of nuanced adjustments. This theoretical capacity to span a larger functional space provides a strong basis for the empirically observed outcome: **WaveFT and SHiRA produce more diverse image generations, as evidenced by the diversity scores in Table 2**.

### 4.1 WHY WAVELETS? A GRADIENT COVERAGE FRAMEWORK

The preceding analysis establishes that both WaveFT and SHiRA produce high-rank updates, escaping LoRA's subspace bottleneck. However, this does not explain *why* WaveFT consistently outperforms SHiRA on vision tasks (Table 2). We introduce a **gradient coverage framework** that explains this phenomenon and predicts when each method excels.

**Gradient Structure Varies Across Tasks.** Our framework hinges on the observation that *gradient sparsity varies significantly across task types*. We define gradient sparsity $\rho$ as the fraction of weight positions receiving informative gradients. In *narrow* fine-tuning tasks (*e.g.* subject personalization), most pretrained weights are already near-optimal, yielding sparse gradients ($\rho \ll 1$). In *broad* tasks (*e.g.* general language understanding), more weights require adaptation, yielding denser gradients.

When gradients are sparse, wavelets' larger receptive fields provide better coverage than SHiRA's point-wise updates. Notably, this advantage does not require spatial coherence; our permutation experiments (Section 5.1) confirm WaveFT maintains its advantage even when input structure is destroyed.

**Gradient Receptive Fields.** Different parameterizations aggregate gradients differently through their **receptive fields**, which determines how effectively they capture gradient signals under varying sparsity conditions.

**Definition 1** (Gradient Receptive Field). For parameter $\theta_i$ in coefficient matrix $C$, the gradient receptive field $\mathcal{R}_i \subseteq [m] \times [n]$ is the set of weight positions whose gradients influence $\theta_i$: $\mathcal{R}_i = \{(u, v) : \partial W_{uv} / \partial \theta_i \neq 0\}$.

The receptive field size depends critically on the transform mapping coefficients to weights:

- **SHiRA** (identity): $|\mathcal{R}_i| = 1$ (each parameter sees only its own gradient)
- **WaveFT** (wavelet, filter size $\kappa$): $|\mathcal{R}_i| = O(\kappa^2)$ (semi-local aggregation)
- **FourierFT** (Fourier): $|\mathcal{R}_i| = mn$ (global aggregation)

**Proposition 1** (Gradient Coverage). *Let $\rho \in (0, 1)$ denote gradient sparsity. When $\rho$ is small and gradient positions are approximately uniformly distributed across the weight matrix, the probability that a parameter with receptive field size $|\mathcal{R}|$ receives gradient signal is approximately: $P(\text{receives gradient}) \approx 1 - (1 - \rho)^{|\mathcal{R}|}$. This approximation holds in expectation over random gradient patterns and random parameter placement. (Proof in Appendix A.1.)*

Table 1 shows coverage for typical fine-tuning sparsity $\rho = 0.1$ with Haar wavelets ($\kappa = 2$). WaveFT achieves $3.4\times$ better gradient coverage than SHiRA.

Table 1: Coverage at $\rho$=0.1.

| Method | $|\mathcal{R}|$ | Coverage | Ratio |
|---|---|---|---|
| SHiRA | 1 | 10% | 1.0× |
| WaveFT | 4 | 34% | 3.4× |
| FourierFT | $mn$ | 100% | 10× |

**The Fourier Failure Mode.** While FourierFT achieves maximal coverage, global receptive fields create two critical problems. First, each Fourier coefficient aggregates gradients from the *entire*

weight matrix: $\partial \mathcal{L} / \partial \hat{C}_{jk} = (mn)^{-1} \sum_{u,v} G_{uv} \cdot e^{2\pi i (ju/m + kv/n)}$.

This causes **destructive interference**: gradients from distant positions, mapped through complex phase factors, can cancel each other even when they individually carry useful signal.

Second, and more fundamentally, sparse FourierFT **overloads each parameter with information** from the entire matrix. A full Fourier representation uses $mn$ coefficients to decompose the signal; sparse FourierFT uses only $p \ll mn$. Each trainable coefficient must therefore encode information aggregated from all positions, without sufficient degrees of freedom to disentangle contributions from different spatial regions. This information bottleneck explains FourierFT's failure on vision tasks where gradients are spatially localized (Table 2).

**Wavelets as the Sweet Spot.** Wavelet bases provide semi-local receptive fields that aggregate gradients from spatially coherent neighborhoods without global interference:

1. **Better coverage than SHiRA:** $3.4\times$ higher probability of receiving gradient signal at $\rho = 0.1$ (Proposition 1)
2. **Constructive aggregation:** Gradients within local $\kappa \times \kappa$ neighborhoods tend to be coherently signed, reinforcing rather than canceling
3. **No information overload:** Each wavelet coefficient encodes information from a bounded spatial region, avoiding the Fourier bottleneck

**Task-Dependent Performance Predictions.** Our framework predicts method performance based on the two gradient properties identified above:

*Cross-domain predictions:* The wavelet advantage emerges primarily when gradients are sparse ($\rho \ll 1$). Spatial structure is not required: wavelets' semi-local receptive fields provide better coverage than SHiRA's point-wise updates regardless of spatial coherence as demonstrated empiricially in the robustness to input permutation ablation in Section 5.1:

- **Vision personalization** (sparse gradients): WaveFT > SHiRA > LoRA
- **NLP tasks** (denser gradients): FourierFT > SHiRA ≈ WaveFT, as global methods benefit when gradient coverage is less critical

*Within-vision predictions:* Even within vision, task granularity matters. Tasks requiring **fine-grained local discrimination** (*e.g.* StanfordCars, FGVC) favor localized methods (WaveFT, SHiRA), while tasks requiring **global pattern recognition** (*e.g.* texture in DTD) may favor global methods (LoRA, FourierFT). We validate these predictions in Section 5.

## 5 EXPERIMENTS

### 5.1 PERSONALIZED TEXT-TO-IMAGE GENERATION

We evaluate WaveFT and SHiRA (Bhardwaj et al., 2024) primarily on personalized text-to-image generation using the SDXL model (Podell et al., 2024) with the 30 DreamBooth instances (Ruiz et al., 2023). Key metrics include DINO (Oquab et al., 2024) and CLIP-I (Radford et al., 2021) similarity for subject fidelity, CLIP-T (Radford et al., 2021) score for prompt alignment, LPIPS (Zhang et al., 2018a) for image diversity, and CMMD (Jayasumana et al., 2024) for distributional similarity to real images. Unless specified otherwise, all methods are configured for a fair comparison with a parameter budget equivalent to LoRA (Hu et al., 2022)

Table 2: PEFT Methods for Personalized Text-to-Image. Best in **bold**. CIs in Appendix Table 9.

| Method | DINO Sim ↑ | CLIP-I Sim ↑ | CLIP-T Score ↑ | LPIPS Div ↑ | CMMD ↓ |
|---|---|---|---|---|---|
| LoRA | 0.463 | 0.640 | 32.39 | 0.309 | 1.275 |
| VeRA | 0.489 | 0.650 | 32.48 | 0.325 | 1.309 |
| AdaLora | 0.468 | 0.642 | 32.34 | 0.306 | 1.274 |
| LoHA | 0.424 | 0.623 | 32.17 | 0.301 | 1.268 |
| FourierFT | 0.215 | 0.518 | 32.32 | 0.250 | **1.173** |
| LoKR | 0.449 | 0.632 | **32.53** | 0.312 | 1.312 |
| SHiRA | 0.467 | 0.645 | 32.09 | 0.342 | 1.254 |
| **WaveFT** | **0.495** | **0.655** | 32.41 | **0.348** | 1.265 |

$r = 1$ ($\approx 1.451$ M trainable parameters for SDXL attention layers). LoRA, WaveFT, and SHiRA all require $\approx 17$ GB of memory during training. WaveFT incurs modest training overhead ($\sim 60\%$ longer than SHiRA due to DWT/IDWT operations) while maintaining identical inference cost through weight merging.

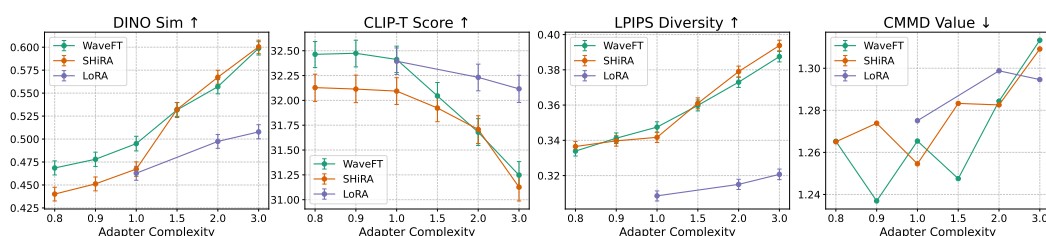

Figure 4: Performance of WaveFT, SHiRA, and LoRA across adapter complexity settings, where the unit complexity (1) is defined as the number of trainable parameters corresponding to the LoRA adapter at rank $r = 1$. WaveFT excels at lower parameter counts for subject fidelity (DINO) while maintaining competitive performance elsewhere. As predicted in Section 4, the advantage of WaveFT over SHiRA is most pronounced when gradients are sparse, and converges in behavior with higher parameter counts. (Also note that at $\hat{r} = 2$ the sparse matrices achieve full rank as shown in Figure 3 in Section 4.)

WaveFT consistently outperforms other PEFT (Mangrulkar et al., 2022) methods in subject fidelity and image diversity while maintaining strong prompt adherence (Table 2). Compared to SHiRA, WaveFT achieves notably higher subject fidelity (DINO: 0.495 vs 0.467, CLIP-I: 0.655 vs 0.645), suggesting that wavelet-domain parameterization provides benefits beyond weight-domain sparsity alone, likely due to improved gradient coverage from semi-local receptive fields.

FourierFT (Gao et al., 2024) exhibits substantially degraded subject fidelity (DINO: 0.215), consistent with our theoretical prediction that global basis functions suffer from destructive interference when gradients are sparse and spatially localized. This poor performance provides empirical support for the gradient coverage framework developed in Section 4.1.

**Parameter Budget Scaling.** We analyze WaveFT against SHiRA across varying parameter budgets (LoRA equivalent ranks 0.8 to 3.0), detailed in Table 7 (appendix) and Figure 4. At lower budgets, WaveFT shows a clear subject fidelity advantage: WaveFT at rank-0.8 equivalent surpasses LoRA at rank-1, demonstrating superior parameter efficiency.

Notably, as the parameter budget increases (Figure 4), the performance gap between WaveFT and SHiRA narrows: at rank-3 equivalent budgets, both methods achieve similar DINO scores ($\sim$0.60). This convergence aligns with our gradient coverage framework: at higher parameter counts, SHiRA's single-position receptive fields eventually achieve sufficient gradient coverage, diminishing the benefit of wavelet aggregation. This observation has practical implications: WaveFT's wavelet-domain parameterization offers the greatest benefit in parameter-constrained regimes.

For prompt fidelity (CLIP-T), WaveFT generally maintains an edge or performs comparably. Image diversity (LPIPS) improves with more parameters for both methods.

**Ablation Studies and Design Choices for WaveFT:** We validated WaveFT's default configuration through several ablations.

- **Initialization:** Zero-initialization of the $p$ trainable parameters in the coefficient matrix $C$ proved robust. Gaussian initialization performed drastically worse (Table 8, appendix), confirming our simpler strategy.
- **Wavelet Family:** Various wavelet families (Coiflets, Daubechies, Symlets) yielded strong, comparable performance (Table 6, appendix). The computationally simpler Daubechies 1 (Haar) was chosen as default due to its robust top-tier results (e.g., Symlet 3, Daubechies 2 in Table 6 show similar performance).
- **Parameter Allocation:** Allocating a fixed $p$ to each layer outperformed allocating parameters proportionally to layer size $(m + n)$ for a similar total budget (Table 8).
- **Location Seed Stability:** WaveFT yields more stable and better results than SHiRA due to it being less sensitive to the random selection of $p$ trainable locations across different seeds (Table 10, appendix).
- **Learned Coefficients Analysis:** The energy levels across wavelet subbands (Fig. 7) did not show clear dominance of any subband, supporting our uniform random selection of trainable coefficients.

- **Robustness to Input Permutation:** To test whether WaveFT's advantage stems from spatial structure, we randomly permuted input token order to attention layers during training and inference. Performance remained largely unchanged (Table 8, "Permuted Input"), with WaveFT still outperforming SHiRA. This key finding demonstrates that WaveFT's advantage arises from improved gradient coverage under sparsity, not from exploiting spatial locality.

**Effect of Output Scaling $\lambda$:** The output scaling factor $\lambda$ in $W = W_0 + \lambda \Delta W$ allows tuning the trade-off between subject fidelity and prompt alignment. For WaveFT, increasing $\lambda \in \{5, ..., 25\}$ generally improved subject fidelity while decreasing prompt alignment (Table 11, Figure 6 in appendix). This provides a controllable mechanism similar to LoRA's $\alpha/r$.

## 5.2 IMAGE CLASSIFICATION TASKS

We further evaluate WaveFT on image classification tasks using a ViT-Base model (Dosovitskiy et al., 2020). Table 3 presents results across multiple datasets: OxfordPets (Parkhi et al., 2012), StanfordCars (Krause et al., 2013), CIFAR-10/100 (Krizhevsky, 2009), DTD (Cimpoi et al., 2014), EuroSAT (Helber et al., 2019), FGVC (Maji et al., 2013), and RESISC45 (Cheng et al., 2017). We compare against Linear Probing (LP), Full Fine-tuning (FF), LoRA, and FourierFT. WaveFT with only 72K trainable parameters achieves the best average performance (78.29%) among PEFT methods at this parameter budget, outperforming LoRA (77.58% at 581K parameters) and FourierFT (77.75%). Notably, WaveFT shows particular strength on fine-grained classification tasks (StanfordCars: 48.12%, FGVC: 31.53%), consistent with our theoretical prediction that localized gradient patterns favor wavelet-domain parameterization.

Table 3: Image classification results on ViT-Base across multiple datasets (median over 5 runs). LP denotes Linear Probing, FF denotes Full Fine-tuning. Best values are in **bold**, second-best in **blue**, third-best in **teal**.

| Method | # Params | OxfordPets | StanfordCars | CIFAR10 | DTD | EuroSAT | FGVC | RESISC45 | CIFAR100 | Avg. |
|---|---|---|---|---|---|---|---|---|---|---|
| LP | – | 90.28 | 25.76 | 96.41 | 69.77 | 88.72 | 17.44 | 74.22 | 84.28 | 68.36 |
| FF | 85.8M | 93.14 | 79.78 | 98.92 | 77.68 | 99.05 | 54.84 | 96.13 | 92.38 | 86.49 |
| LoRA | 581K | 93.19 | 45.38 | 98.78 | 74.95 | 98.44 | 25.16 | 92.70 | 92.02 | 77.58 |
| FourierFT | 72K | 93.21 | 46.11 | 98.58 | 75.09 | 98.29 | 27.51 | 91.97 | 91.20 | 77.75 |
| SHiRA | 72K | 91.50 | 47.48 | 98.56 | 72.66 | 98.93 | 31.32 | 92.84 | 90.85 | 78.02 |
| **WaveFT (ours)** | **72K** | 91.82 | 48.12 | 98.61 | 73.24 | 98.96 | 31.53 | 92.98 | 91.09 | 78.29 |

**Additional Experiments.** We provide empirical validation of Lemma 3 (block-sparse interpolation capacity) in Appendix A.3.1, demonstrating that sparse matrices can perfectly interpolate arbitrary input-output mappings when sufficient parameters are appropriately distributed. We also evaluate WaveFT on the GLUE benchmark (Wang et al., 2018) for language understanding (Appendix A.3.3), where WaveFT and SHiRA perform comparably (83.9 vs 84.2 average), both slightly below FourierFT (85.0). This aligns with our theoretical predictions: NLP fine-tuning involves denser, more distributed gradient patterns, reducing the locality advantage of wavelets and allowing global methods like FourierFT to effectively aggregate gradients.

## 6 CONCLUSION

We introduced Wavelet Fine-Tuning (WaveFT), a parameter-efficient fine-tuning method that learns sparse updates in the wavelet domain. Our theoretical analysis establishes two key insights: (i) sparse methods produce high-rank updates, escaping LoRA's subspace bottleneck and yielding higher representational capacity that translates to more diverse outputs, and (ii) wavelet bases provide semi-local gradient aggregation, achieving better coverage than weight-space sparsity (SHiRA) without the destructive interference of global Fourier bases (FourierFT).

Our experiments span text-to-image generation (SDXL), image classification (ViT-Base), and language understanding (GLUE). WaveFT achieves state-of-the-art results among PEFT methods on vision tasks, with particular strength on fine-grained classification and personalized generation where gradients are sparse and localized. Ablation studies confirm that WaveFT's advantage stems from gradient coverage rather than spatial structure, while also demonstrating superior stability across random seeds and consistent performance across wavelet families.

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

# A   APPENDIX

**Appendix Contents**

## A.1 PROOFS AND THEORETICAL ANALYSIS

### A.1.1 PROOF OF SUBSPACE BOTTLENECK LEMMA

**Lemma** (Subspace Bottleneck of LoRA). *For a rank-$r$ adapter update matrix of the form $\Delta W = BA^\top$, where $B \in \mathbb{R}^{m \times r}$ and $A \in \mathbb{R}^{n \times r}$, the following properties hold:*

1. *The image (column space) of $\Delta W$ is contained within the span of the columns of $B$:*

$$\text{im}(\Delta W) = \{\Delta W x \mid x \in \mathbb{R}^n\} \subseteq \text{span}(\textit{columns of } B).$$

2. *The kernel (null space) of $\Delta W$ contains the orthogonal complement of the span of the columns of $A$:*

$$\ker(\Delta W) = \{x \in \mathbb{R}^n \mid \Delta W x = 0\} \supseteq (\text{span}(\textit{columns of } A))^\perp = \ker(A^\top).$$

*Consequently, any update $\Delta W$ achieved through such a factorization can only modify the network's activations within the $r$-dimensional subspace spanned by the columns of $B$. Directions orthogonal to the columns of $A$ in the input space are mapped to zero.*

*Proof.* 1. For any $x$, $\Delta W x = B(A^\top x)$ is a linear combination of $B$'s columns, so $\text{im}(\Delta W) \subseteq \text{span}(B)$. 2. If $x \in \ker(A^\top)$ then $A^\top x = 0$, so $\Delta W x = B0 = 0$, hence $\ker(A^\top) \subseteq \ker(\Delta W)$. This completes the proof of the bottleneck. $\square$

### A.1.2 PROOF OF BLOCK-SPARSE INTERPOLATION CAPACITY

**Lemma** (Block-Sparse Interpolation Capacity). *Let $W_0 \in \mathbb{R}^{m \times n}$ be any fixed matrix. Let*

$$\{x^{(1)}, \ldots, x^{(k)}\} \subset \mathbb{R}^n$$

*be linearly independent, and let arbitrary targets $\{y^{(1)}, \ldots, y^{(k)}\} \subset \mathbb{R}^m$ be given. Set*

$$X = \begin{bmatrix} x^{(1)} & \cdots & x^{(k)} \end{bmatrix} \in \mathbb{R}^{n \times k}, \quad Z = \begin{bmatrix} y^{(1)} - W_0 x^{(1)} & \cdots & y^{(k)} - W_0 x^{(k)} \end{bmatrix} \in \mathbb{R}^{m \times k}.$$

*Let $S \subset [m] \times [n]$ be a fixed sparse support pattern, and define*

$$R = \{\, i \in [m] \mid Z_{i,:} \neq 0 \,\}, \quad S_i = \{\, j \in [n] \mid (i, j) \in S \,\}.$$

*Assume:*

1. $\text{rank}(X) = k$.

2. *There exists a single index set*

$$C = \{c_1, \ldots, c_k\} \subset [n]$$

*such that $X_{C,:} \in \mathbb{R}^{k \times k}$ is invertible and $C \subset S_i$ for every $i \in R$.*

*Then one can construct $\Delta W \in \mathbb{R}^{m \times n}$ with*

1. $\text{supp}(\Delta W) \subseteq S$.

2. $(W_0 + \Delta W)\, x^{(l)} = y^{(l)}$ *for all $l = 1, \ldots, k$.*

3. $\text{rank}(\Delta W) = \text{rank}(Z_R)$, *where $Z_R$ is the submatrix of $Z$ restricted to rows in $R$.*

*Proof.* **Step 1: Existence of an invertible block.** Since $\text{rank}(X) = k$, there exists at least one $k$-subset $C \subset [n]$ for which the submatrix $X_{C,:}$ is nonsingular. By hypothesis (2), we choose such a $C$ and furthermore have $C \subset S_i$ for every $i \in R$.

**Step 2: Construction of $\Delta W$.** Define $\Delta W$ row-wise by

$$(\Delta W)_{i,j} = \begin{cases} \left(Z_{i,:}\, (X_{C,:})^{-1}\right)_r, & i \in R, \ j = c_r \in C, \\ 0, & \text{otherwise.} \end{cases}$$

Equivalently, for each $i \in R$ the row $(\Delta W)_{i,:}$ has its only potentially nonzero entries in columns $C$, given by the $1 \times k$ vector $Z_{i,:}(X_{C,:})^{-1}$. For $i \notin R$, set the $i$-th row to zero.

**Step 3: Verification of properties.**

*(i) Sparsity.* By construction, the only nonzero entries of row $i \in R$ lie in columns $C \subset S_i$, and rows $i \notin R$ are entirely zero. Hence $\mathrm{supp}(\Delta W) \subseteq S$.

*(ii) Exact interpolation $\Delta W X = Z$.* Fix any row $i$:

- If $i \notin R$, then $Z_{i,:} = 0$ and $(\Delta W)_{i,:} = 0$, so $(\Delta W)_{i,:}X = 0 = Z_{i,:}$.

- If $i \in R$, only columns in $C$ contribute:
$$(\Delta W)_{i,:}X = \left(Z_{i,:}(X_{C,:})^{-1}\right)X_{C,:} = Z_{i,:}\,,$$
since $X_{C,:}$ is invertible. Thus $\Delta W X = Z$, and $(W_0 + \Delta W)x^{(l)} = W_0 x^{(l)} + Z_{:,l} = y^{(l)}$ for each $l$.

*(iii) Rank lower bound.* Let $\Delta W_R$ and $Z_R$ be the submatrices restricted to rows in $R$. Observe that $\Delta W_R = Z_R (X_{C,:})^{-1} E$, where $E \in \mathbb{R}^{k \times n}$ is the embedding that places the $k$ columns into positions $C$. Note:
$$\mathrm{rank}(\Delta W_R) = \mathrm{rank}\left(Z_R (X_{C,:})^{-1} E\right) = \mathrm{rank}(Z_R),$$
because left-multiplication by the invertible $(X_{C,:})^{-1}$ and the column-embedding $E$ both preserve the row-rank. Rows outside $R$ of $\Delta W$ are zero, so $\mathrm{rank}(\Delta W) = \mathrm{rank}(\Delta W_R) = \mathrm{rank}(Z_R)$.

This completes the proof. $\square$

### A.1.3 PROOF OF GRADIENT COVERAGE PROPOSITION

**Proposition** (Gradient Coverage). *Let $\rho \in (0, 1)$ denote gradient sparsity. When $\rho$ is small and gradient positions are approximately uniformly distributed across the weight matrix, the probability that a parameter with receptive field size $|\mathcal{R}|$ receives gradient signal is approximately: $P(\text{receives gradient}) \approx 1 - (1 - \rho)^{|\mathcal{R}|}$. This approximation holds in expectation over random gradient patterns and random parameter placement.*

*Proof.* Consider a trainable parameter $\theta_i$ with gradient receptive field $\mathcal{R}_i$ of size $|\mathcal{R}|$. The parameter receives gradient signal if at least one position $(u, v) \in \mathcal{R}_i$ contains a significant gradient.

Under the assumption that gradient positions are approximately uniformly distributed across the $mn$ weight positions with sparsity $\rho$, each position independently contains significant gradient with probability $\rho$. The parameter receives *no* gradient signal only if all positions in its receptive field miss:
$$P(\text{no gradient}) = \prod_{(u,v) \in \mathcal{R}_i} P(|G_{uv}| \leq \tau) = (1 - \rho)^{|\mathcal{R}|}$$

Therefore:
$$P(\text{receives gradient}) = 1 - (1 - \rho)^{|\mathcal{R}|}$$

This approximation is most accurate when: (i) $\rho$ is small, so gradient positions are sparse and approximately independent; (ii) the receptive field $\mathcal{R}_i$ is small relative to $mn$, avoiding boundary effects; and (iii) gradient positions do not exhibit strong spatial clustering within receptive field scales.

For the comparison in Table 1, we set $\rho = 0.1$ (typical for narrow fine-tuning tasks). With Haar wavelets ($\kappa = 2$), WaveFT has $|\mathcal{R}| = 4$, yielding coverage $1 - 0.9^4 \approx 0.344$, compared to SHiRA's $|\mathcal{R}| = 1$ with coverage $0.1$. This $3.4\times$ improvement explains WaveFT's advantage when gradients are sparse and spatially structured. $\square$

## A.2 EXPERIMENTAL SETUP AND METHODOLOGY

### A.2.1 PERSONALIZED TEXT-TO-IMAGE GENERATION

**Experimental Setup.** **Model and Task:** All experiments are conducted using the Stable Diffusion XL (SDXL) 1.0 base model Podell et al. (2024). We focus on the task of personalized text-to-image generation, employing the methodology as proposed by DreamBooth Ruiz et al. (2023) and training only the corresponding PEFT adapters while freezing the pretrained weights.

**Dataset:** We utilize the full set of 30 diverse instances from the DreamBooth benchmark for all main experiments. This dataset encompasses a variety of live subjects and objects. The corresponding real images provided for each instance are used as references for subject fidelity evaluation metrics (DINO, CLIP-I).

**Training Details:** For each of the 30 instances and every PEFT method evaluated, fine-tuning is applied exclusively to the parameters of the attention layers (specifically, the key, query, value, and output projection matrices within all attention blocks) of the SDXL UNet. The text encoder and all other components of the UNet remain frozen, adhering to common practices for efficient personalization. We employ the AdamW optimizer Loshchilov & Hutter (2019) with a constant learning rate of $1 \times 10^{-4}$ and train for 500 steps. A per-device batch size of 1 is used, with gradient accumulation over 4 steps, resulting in an effective batch size of 4. All training is performed at the standard SDXL resolution of $1024 \times 1024$ pixels.

**Parameter Budget:** For our main comparisons (Table 8), the number of trainable parameters $p$ for WaveFT and SHiRA Bhardwaj et al. (2024) is configured to closely match the parameter count of LoRA Hu et al. (2022) with rank $r = 1$. For the targeted attention layers in SDXL, this amounts to approximately 1.451 million trainable parameters. We refer to this configuration as the 'rank-1 equivalent' budget. Parameter counts for all methods are calculated based on the trainable weights within these specified UNet attention layers. For experiments analyzing the effect of varying parameter budgets (e.g., Table 7), $p$ is adjusted accordingly.

We utilize implementations from the Hugging Face PEFT library Mangrulkar et al. (2022) using their standard configurations where applicable, to ensure reproducibility and fairness. Our WaveFT and SHiRA implementations are designed for compatibility.

**Other Hyperparameters:** Our proposed method WaveFT and SHiRA, by default, utilize zero-initialized trainable parameters, an adapter output scaling factor $\lambda = 25$, and a fixed number of $p$ parameters per adapted layer. For WaveFT, the Daubechies 1 (Haar) wavelet is the default. Experiments were run in bf16 precision with equal parameter budget, with exceptions for LoHA (approx. 2.9M parameters due to its architecture) and FourierFT (fp32 due to library limitations), whose results should be considered in this context.

**$\lambda$-equivalent parameters for baseline adapters:** For all baseline methods, we conducted comprehensive hyperparameter searches to determine optimal configurations that ensure fair comparison. For LoRA, following the conventions in diffusers von Platen et al. (2022) for the SDXL model, we set $\alpha = r$ (where $r$ is the rank). For other methods, we determined the following optimal configurations:

- **VeRA**: Learning rate of $3.2 \times 10^{-3}$ (32 times the base learning rate)
- **AdaLoRA**: $\alpha = 32$
- **LoHA**: $\alpha = 64$
- **FourierFT**: Scaling factor scale $= 64$
- **LoKR**: $\alpha = 192$

These hyperparameter values were determined through ablation studies to ensure each method performs optimally within its parameter budget constraints. For all methods, we utilized implementations from the Hugging Face PEFT library with their standard configurations where applicable, modified only by the parameters specified above.

**Evaluation Protocol:** For each of the 30 instances, we generate 4 images for each of the 25 standard prompts provided by the DreamBooth benchmark. This results in 100 generated images per instance (3000 images per method in total across all instances) for quantitative evaluation. All images are generated using a fixed set of seeds for comparability across methods.

**Evaluation Metrics.**   We assess the performance of each PEFT method using a comprehensive suite of metrics targeting different facets of personalized image generation quality:

- **DINO Score (Subject Fidelity):** Measures the average cosine similarity between DINOv2 Oquab et al. (2024) ViT-B/14 (`facebook/dinov2-base`) CLS token embeddings of generated images and the average DINOv2 CLS token embedding of the corresponding real images for the specific instance subject. Higher scores indicate better visual resemblance to the target subject's identity and key features.

- **CLIP-I Score (Subject Fidelity):** Calculates the average cosine similarity between CLIP Radford et al. (2021) ViT-B/32 (`openai/clip-vit-base-patch32`) image embeddings (pooler output) of generated images and the average CLIP image embedding of the real images for the instance. This offers another perspective on subject fidelity through CLIP's image feature space. Higher scores are better.

- **CLIP-T Score (Prompt Fidelity):**   Computes the average CLIP score (using `openai/clip-vit-base-patch32`) between the generated images and their corresponding input text prompts. This metric evaluates how well the generated image aligns with the textual description. Higher scores indicate better prompt adherence.

- **Diversity (DIV) Score (Intra-Prompt Dissimilarity):** Assesses the diversity of images generated for the same prompt. We calculate the average pairwise Learned Perceptual Image Patch Similarity (LPIPS) Zhang et al. (2018b) (using the TorchMetrics Detlefsen et al. (2022) implementation with VGG weights and input normalization) between the 4 images generated for each prompt (resulting in $Comb(4, 2) = 6$ pairs). This is then averaged across all prompts and instances following the DreamBooth protocol. Then this average is subtracted from 1 to measure diversity rather than similarity.

- **CMMD (Distributional Similarity to Real Images):** We compute the CLIP-based Maximum Mean Discrepancy (CMMD) Jayasumana et al. (2024) using CLIP ViT-B/32 image embeddings. This metric compares the distribution of embeddings from all 3000 generated images (across all instances and prompts for a given method) against the distribution of embeddings from a large reference set of real images (COCO-30k Lin et al. (2014)). **Lower CMMD scores are better**, indicating that the overall distribution of generated images is closer to that of natural images.

For DINO, CLIP-I, CLIP-T, and DIV scores, we report the mean over the 30 instances. Confidence intervals (95%) are provided for all metrics, calculated via bootstrapping with 10,000 iterations over the 30 instances.

**Computational Complexity.**   All of the experiments above are done in 46gb NVIDIA A40 GPU's. LoRA takes around 20 minutes to train for a single instance, SHiRA also takes around 22 minutes and WaveFT takes about 34 minutes.

A significant advantage of WaveFT, which is shared with LoRA, is their inference efficiency. Once trained, the learned adapter update $\Delta W$ can be merged with the base model weights $W_0$, thereby incurring no additional computational latency during inference compared to the original model, as discussed in Section 3.

During training, SHiRA is computationally efficient as it only requires updating $p$ sparse parameters directly in the coefficient matrix $C$ and involves sparse matrix operations. WaveFT introduces an additional computational step compared to SHiRA due to the Inverse Discrete Wavelet Transform (IDWT) applied to the sparse coefficient matrix $C$ to form $\Delta W_{\text{WaveFT}}$, and the corresponding Discrete Wavelet Transform (DWT) in the backward pass for gradient computation. The complexity of these transforms depends on the chosen wavelet (e.g., Daubechies 1/Haar) and implementation, but is typically efficient, especially considering that $C$ is sparse. Zero-padding is applied as needed if matrix dimensions $(m, n)$ are not ideal for standard 2D DWT/IDWT algorithms.

It is important to note that all these PEFT methods (WaveFT, SHiRA, and LoRA) offer substantial reductions in both the number of trainable parameters and overall training time compared to full fine-tuning of the large-scale SDXL model.

In summary, while WaveFT incurs a modest computational overhead during training compared to the direct sparse updates of SHiRA due to the wavelet transforms, WaveFT maintains the crucial advantage of zero latency overhead at inference time.

### A.3 ADDITIONAL EXPERIMENTAL RESULTS

#### A.3.1 EMPIRICAL VALIDATION OF BLOCK-SPARSE INTERPOLATION CAPACITY

To illustrate the practical implications of Lemma 3, we conducted an experiment involving mapping $k = 5$ random input vectors to $k = 5$ random output vectors in a high-dimensional space. This setup creates a representative interpolation problem that aligns with the conditions in Lemma 3, where we set the dimensionality to $784 \times 784$ and tested varying numbers of trainable parameters.

**Experimental Setup.** We implemented a sparse matrix model (SHiRA Bhardwaj et al. (2024)) which constructs a weight matrix with trainable parameters randomly selected from the total $784^2$ possible positions. The model was trained to map $5$ random input vectors of dimension $784$ to $5$ random output vectors of the same dimension, all sampled from a normal distribution. The remaining elements were fixed at zero throughout training.

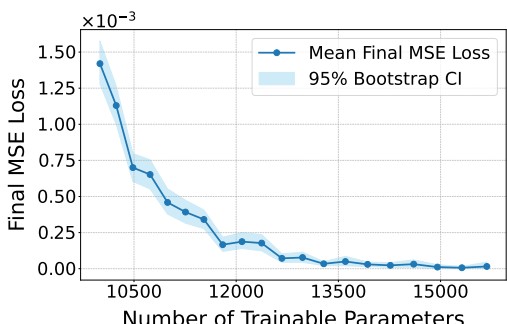

Figure 5: Final training loss as a function of trainable parameters $p$. Loss converges to zero at $p \approx 15,680$, validating block-sparse interpolation capacity.

We used the Mean Squared Error (MSE) as the loss function and Adam optimizer Kingma & Ba (2015) with an initial learning rate of $0.01$. A step scheduler reduced the learning rate by a factor of $\gamma = 0.75$ every $500$ epochs. The model was trained for a total of $5,000$ epochs to ensure convergence. We systematically varied the number of trainable parameters to determine the minimum count required for successful interpolation.

**Performance Evaluation.** The empirical results shown in Figure 5 strongly support our theoretical analysis. At approximately $15,680$ trainable parameters, the training loss converges to zero with high confidence, demonstrating that the sparse model successfully learns to map the input vectors to their target outputs with high precision.

This confirms that when sufficient trainable parameters are appropriately distributed through random selection, the sparse matrix can perfectly interpolate between arbitrary input-output pairs, validating the practical implications of Lemma 3. This result is particularly relevant for understanding the effectiveness of SHiRA, which leverages sparse parameterization to achieve high representational capacity with a small number of trainable parameters.

### A.3.2 Image Classification Results with Statistical Details

Table 4: Image classification results on ViT-Base (Dosovitskiy et al., 2020) with median$_{\pm\text{std}}$ over 5 runs. LP = Linear Probing, FF = Full Fine-tuning.

| Method | # Params | OxfordPets | StanfordCars | CIFAR10 | DTD | EuroSAT | FGVC | RESISC45 | CIFAR100 |
|---|---|---|---|---|---|---|---|---|---|
| LP | – | $90.28_{\pm0.43}$ | $25.76_{\pm0.28}$ | $96.41_{\pm0.02}$ | $69.77_{\pm0.67}$ | $88.72_{\pm0.13}$ | $17.44_{\pm0.43}$ | $74.22_{\pm0.10}$ | $84.28_{\pm0.11}$ |
| FF | 85.8M | $93.14_{\pm0.40}$ | $\mathbf{79.78}_{\pm1.15}$ | $\mathbf{98.92}_{\pm0.05}$ | $\mathbf{77.68}_{\pm1.21}$ | $\mathbf{99.05}_{\pm0.09}$ | $\mathbf{54.84}_{\pm1.23}$ | $\mathbf{96.13}_{\pm0.13}$ | $\mathbf{92.38}_{\pm0.13}$ |
| LoRA | 581K | $93.19_{\pm0.36}$ | $45.38_{\pm0.41}$ | $98.78_{\pm0.05}$ | $74.95_{\pm0.40}$ | $98.44_{\pm0.15}$ | $25.16_{\pm0.16}$ | $92.70_{\pm0.18}$ | $92.02_{\pm0.12}$ |
| FourierFT | 72K | $\mathbf{93.21}_{\pm0.26}$ | $46.11_{\pm0.24}$ | $98.58_{\pm0.07}$ | $75.09_{\pm0.37}$ | $98.29_{\pm0.04}$ | $27.51_{\pm0.64}$ | $91.97_{\pm0.31}$ | $91.20_{\pm0.14}$ |
| SHiRA | 72K | $91.50_{\pm0.19}$ | $47.48_{\pm0.55}$ | $98.56_{\pm0.06}$ | $72.66_{\pm0.39}$ | $98.93_{\pm0.14}$ | $31.32_{\pm0.71}$ | $92.84_{\pm0.28}$ | $90.85_{\pm0.05}$ |
| **WaveFT** | **72K** | $91.82_{\pm0.15}$ | $48.12_{\pm0.79}$ | $98.61_{\pm0.05}$ | $73.24_{\pm0.69}$ | $98.96_{\pm0.10}$ | $31.53_{\pm1.84}$ | $92.98_{\pm0.15}$ | $91.09_{\pm0.12}$ |

**Training Time.** Across the 8 image classification tasks, WaveFT requires approximately 80 minutes total training time ($\sim$10 min per task on average), while SHiRA requires approximately 78 minutes ($\sim$2% faster). This modest overhead stems from the DWT/IDWT operations in WaveFT, consistent with observations on SDXL (Section A.2.1).

### A.3.3 Language Understanding: GLUE Benchmark

We evaluate WaveFT on the GLUE benchmark (Wang et al., 2018) using RoBERTa-base (Liu et al., 2019) to assess performance on natural language understanding tasks. Table 5 presents results across six tasks: SST-2, MRPC, CoLA, QNLI, RTE, and STS-B. We compare against several PEFT baselines: BitFit (Zaken et al., 2021), Adapter (Adpt$^D$) (Houlsby et al., 2019), LoRA (Hu et al., 2022), AdaLoRA (Zhang et al., 2023), DyLoRA (Valipour et al., 2022), and FourierFT (Gao et al., 2024).

Table 5: Performance comparison of WaveFT against other parameter-efficient fine-tuning methods on the GLUE benchmark using RoBERTa-base. We report accuracy for all tasks except CoLA (MCC) and STS-B (Pearson). Best in **bold**, second-best in blue, third-best in teal.

| Method | # Params | SST-2 (Acc.) | MRPC (Acc.) | CoLA (MCC) | QNLI (Acc.) | RTE (Acc.) | STS-B (PCC) | Avg. |
|---|---|---|---|---|---|---|---|---|
| FF | 125M | 94.8 | 90.2 | 63.6 | 92.8 | 78.7 | 91.2 | 85.2 |
| BitFit | 0.1M | 93.7 | **92.7** | 62.0 | 91.8 | **81.5** | 90.8 | **85.4** |
| Adpt$^D$ | 0.3M | $94.2_{\pm0.1}$ | $88.5_{\pm1.1}$ | $60.8_{\pm0.4}$ | $93.1_{\pm0.1}$ | $71.5_{\pm2.7}$ | $89.7_{\pm0.3}$ | 83.0 |
| Adpt$^D$ | 0.9M | $94.7_{\pm0.3}$ | $88.4_{\pm0.1}$ | $62.6_{\pm0.9}$ | $93.0_{\pm0.2}$ | $75.9_{\pm2.2}$ | $90.3_{\pm0.1}$ | 84.2 |
| LoRA | 0.3M | $95.1_{\pm0.2}$ | $89.7_{\pm0.7}$ | $63.4_{\pm1.2}$ | $93.3_{\pm0.3}$ | $78.4_{\pm0.8}$ | $91.5_{\pm0.2}$ | 85.2 |
| AdaLoRA | 0.3M | $94.5_{\pm0.2}$ | $88.7_{\pm0.5}$ | $62.0_{\pm0.6}$ | $93.1_{\pm0.2}$ | $81.0_{\pm0.6}$ | $90.5_{\pm0.2}$ | 85.0 |
| DyLoRA | 0.3M | $94.3_{\pm0.5}$ | $89.5_{\pm0.5}$ | $61.1_{\pm0.3}$ | $92.2_{\pm0.5}$ | $78.7_{\pm0.7}$ | $91.1_{\pm0.6}$ | 84.5 |
| FourierFT | 0.024M | $94.2_{\pm0.3}$ | $90.0_{\pm0.8}$ | $63.8_{\pm1.6}$ | $92.2_{\pm0.1}$ | $79.1_{\pm0.5}$ | $90.8_{\pm0.2}$ | 85.0 |
| SHiRA | 0.024M | $94.0_{\pm0.4}$ | $90.8_{\pm0.7}$ | $59.8_{\pm0.6}$ | $90.9_{\pm0.1}$ | $79.8_{\pm0.7}$ | $89.6_{\pm0.2}$ | 84.2 |
| **WaveFT** | 0.024M | $94.3_{\pm0.1}$ | $90.6_{\pm0.5}$ | $59.6_{\pm1.1}$ | $91.1_{\pm0.2}$ | $78.7_{\pm0.7}$ | $89.3_{\pm0.1}$ | 83.9 |

On GLUE, WaveFT and SHiRA perform comparably (83.9 vs 84.2 average), both slightly below LoRA (85.2) and FourierFT (85.0). This aligns with our theoretical predictions: NLP tasks lack the spatial structure that gives wavelets their advantage, and the denser gradient patterns favor global methods like FourierFT. The comparable performance of WaveFT and SHiRA on GLUE, contrasted with WaveFT's clear advantage on vision tasks, validates our gradient coverage framework's task-dependent predictions.

**Training Time.** Across the 6 GLUE tasks, WaveFT requires approximately 168 minutes total training time, while SHiRA requires approximately 92 minutes ($\sim$45% faster). This larger overhead compared to vision tasks reflects the DWT/IDWT operations being applied more frequently due to higher training throughput in NLP.

### A.3.4 Ablation Tables

Table 6: Evaluation Summary for Wavelet Families (Coiflet, Symlet, Debauchies) Ordered by Name. Confidence intervals are shown below the mean value as [*lower difference, +upper difference*].

| Configuration Name | DINO Sim ↑ | CLIP-I Sim ↑ | CLIP-T Score ↑ | LPIPS Diversity ↑ | CMMD Value ↓ |
|---|---|---|---|---|---|
| Debauchies 1 | 0.4950 *[-0.0079, +0.0080]* | 0.6545 *[-0.0043, +0.0043]* | 32.4121 *[-0.1317, +0.1339]* | 0.3475 *[-0.0030, +0.0029]* | 1.265 |
| Debauchies 2 | 0.4942 *[-0.0081, +0.0076]* | 0.6544 *[-0.0042, +0.0042]* | 32.3726 *[-0.1312, +0.1297]* | 0.3420 *[-0.0030, +0.0029]* | 1.300 |
| Debauchies 3 | 0.4930 *[-0.0082, +0.0078]* | 0.6531 *[-0.0043, +0.0043]* | 32.4174 *[-0.1343, +0.1361]* | 0.3433 *[-0.0029, +0.0030]* | 1.312 |
| Coiflet 1 | 0.4893 *[-0.0077, +0.0079]* | 0.6513 *[-0.0041, +0.0041]* | 32.2810 *[-0.1329, +0.1352]* | 0.3422 *[-0.0029, +0.0029]* | 1.279 |
| Coiflet 2 | 0.4926 *[-0.0079, +0.0079]* | 0.6546 *[-0.0044, +0.0043]* | 32.2956 *[-0.1339, +0.1358]* | 0.3456 *[-0.0029, +0.0028]* | 1.306 |
| Symlet 2 | 0.4930 *[-0.0078, +0.0079]* | 0.6547 *[-0.0044, +0.0042]* | 32.3512 *[-0.1349, +0.1335]* | 0.3422 *[-0.0030, +0.0030]* | 1.303 |
| Symlet 3 | 0.4950 *[-0.0077, +0.0079]* | 0.6548 *[-0.0044, +0.0041]* | 32.3891 *[-0.1347, +0.1351]* | 0.3453 *[-0.0029, +0.0030]* | 1.321 |
| Symlet 4 | 0.4938 *[-0.0081, +0.0078]* | 0.6534 *[-0.0041, +0.0043]* | 32.3615 *[-0.1369, +0.1328]* | 0.3463 *[-0.0031, +0.0029]* | 1.278 |

Table 7: Evaluation Summary for LoRA, SHiRA, and WaveFT Configurations Sorted by Rank. Confidence intervals are shown below the mean value as [*lower difference, +upper difference*].

| Configuration Name | DINO Sim ↑ | CLIP-I Sim ↑ | CLIP-T Score ↑ | LPIPS Diversity ↑ | CMMD Value ↓ |
|---|---|---|---|---|---|
| WaveFT (rank=0.8) | 0.4685 *[-0.0076, +0.0077]* | 0.6418 *[-0.0042, +0.0041]* | 32.4637 *[-0.1333, +0.1290]* | 0.3339 *[-0.0028, +0.0028]* | 1.265 |
| SHiRA (rank=0.8) | 0.4401 *[-0.0075, +0.0074]* | 0.6320 *[-0.0041, +0.0042]* | 32.1286 *[-0.1387, +0.1348]* | 0.3365 *[-0.0029, +0.0028]* | 1.265 |
| SHiRA (rank=0.9) | 0.4512 *[-0.0076, +0.0075]* | 0.6389 *[-0.0043, +0.0041]* | 32.1140 *[-0.1368, +0.1399]* | 0.3397 *[-0.0029, +0.0030]* | 1.273 |
| WaveFT (rank=0.9) | 0.4780 *[-0.0078, +0.0077]* | 0.6449 *[-0.0043, +0.0042]* | 32.4744 *[-0.1369, +0.1315]* | 0.3412 *[-0.0030, +0.0030]* | 1.236 |
| LoRA | 0.4628 *[-0.0077, +0.0075]* | 0.6400 *[-0.0042, +0.0041]* | 32.3946 *[-0.1334, +0.1336]* | 0.3085 *[-0.0028, +0.0029]* | 1.275 |
| SHiRA | 0.4673 *[-0.0079, +0.0078]* | 0.6451 *[-0.0041, +0.0041]* | 32.0934 *[-0.1343, +0.1350]* | 0.3417 *[-0.0029, +0.0029]* | 1.254 |
| WaveFT | 0.4950 *[-0.0079, +0.0080]* | 0.6545 *[-0.0043, +0.0043]* | 32.4121 *[-0.1317, +0.1339]* | 0.3475 *[-0.0030, +0.0029]* | 1.265 |
| SHiRA (rank=1.5) | 0.5322 *[-0.0077, +0.0076]* | 0.6744 *[-0.0041, +0.0040]* | 31.9234 *[-0.1384, +0.1375]* | 0.3610 *[-0.0031, +0.0031]* | 1.283 |
| WaveFT (rank=1.5) | 0.5317 *[-0.0082, +0.0080]* | 0.6734 *[-0.0043, +0.0042]* | 32.0445 *[-0.1382, +0.1356]* | 0.3598 *[-0.0031, +0.0031]* | 1.247 |
| LoRA (rank=2) | 0.4974 *[-0.0075, +0.0075]* | 0.6553 *[-0.0040, +0.0040]* | 32.2320 *[-0.1357, +0.1330]* | 0.3150 *[-0.0029, +0.0029]* | 1.298 |
| SHiRA (rank=2) | 0.5673 *[-0.0076, +0.0076]* | 0.6918 *[-0.0039, +0.0039]* | 31.7078 *[-0.1425, +0.1375]* | 0.3790 *[-0.0031, +0.0032]* | 1.282 |
| WaveFT (rank=2) | 0.5570 *[-0.0078, +0.0077]* | 0.6881 *[-0.0042, +0.0042]* | 31.6796 *[-0.1334, +0.1361]* | 0.3730 *[-0.0029, +0.0030]* | 1.284 |
| LoRA (rank=3) | 0.5078 *[-0.0075, +0.0078]* | 0.6622 *[-0.0041, +0.0040]* | 32.1163 *[-0.1379, +0.1364]* | 0.3207 *[-0.0030, +0.0030]* | 1.294 |
| SHiRA (rank=3) | 0.6004 *[-0.0075, +0.0072]* | 0.7078 *[-0.0039, +0.0038]* | 31.1268 *[-0.1396, +0.1384]* | 0.3938 *[-0.0030, +0.0030]* | 1.309 |
| WaveFT (rank=3) | 0.5988 *[-0.0073, +0.0072]* | 0.7041 *[-0.0039, +0.0039]* | 31.2469 *[-0.1350, +0.1362]* | 0.3875 *[-0.0029, +0.0030]* | 1.313 |

Table 8: Evaluation Summary for Ablations on WaveFT. Confidence intervals are shown below the mean value as [*-lower difference, +upper difference*].

| Configuration Name | DINO Sim ↑ | CLIP-I Sim ↑ | CLIP-T Score ↑ | LPIPS Diversity ↑ | CMMD Value ↓ |
|---|---|---|---|---|---|
| Base Version | 0.4950 *[-0.0079, +0.0080]* | 0.6545 *[-0.0043, +0.0043]* | 32.4121 *[-0.1317, +0.1339]* | 0.3475 *[-0.0030, +0.0029]* | 1.265 |
| Proportional Parameter Allocation | 0.4729 *[-0.0079, +0.0076]* | 0.6436 *[-0.0042, +0.0042]* | 32.4038 *[-0.1365, +0.1341]* | 0.3302 *[-0.0029, +0.0028]* | 1.255 |
| Permutated Input Embedding Experiment | 0.4871 *[-0.0080, +0.0078]* | 0.6519 *[-0.0042, +0.0041]* | 32.2815 *[-0.1325, +0.1277]* | 0.3440 *[-0.0030, +0.0030]* | 1.271 |
| Gaussian Initialization | 0.0130 *[-0.0014, +0.0013]* | 0.3315 *[-0.0017, +0.0016]* | 20.1346 *[-0.0923, +0.0931]* | 0.3962 *[-0.0013, +0.0013]* | 3.707 |

Table 9: Evaluation Summary for Different Methods with 95% CIs. Confidence intervals are shown below the mean value as [*-lower difference, +upper difference*].

| Configuration Name | DINO Sim ↑ | CLIP-I Sim ↑ | CLIP-T Score ↑ | LPIPS Diversity ↑ | CMMD Value ↓ |
|---|---|---|---|---|---|
| LoRA | 0.4628 *[-0.0077, +0.0075]* | 0.6400 *[-0.0042, +0.0041]* | 32.3946 *[-0.1334, +0.1336]* | 0.3085 *[-0.0028, +0.0029]* | 1.275 |
| SHiRA | 0.4673 *[-0.0079, +0.0078]* | 0.6451 *[-0.0041, +0.0041]* | 32.0934 *[-0.1343, +0.1350]* | 0.3417 *[-0.0029, +0.0029]* | 1.254 |
| WaveFT | 0.4950 *[-0.0079, +0.0080]* | 0.6545 *[-0.0043, +0.0043]* | 32.4121 *[-0.1317, +0.1339]* | 0.3475 *[-0.0030, +0.0029]* | 1.265 |
| VeRA | 0.4889 *[-0.0075, +0.0078]* | 0.6496 *[-0.0043, +0.0041]* | 32.4818 *[-0.1315, +0.1336]* | 0.3246 *[-0.0029, +0.0028]* | 1.309 |
| AdaLora | 0.4676 *[-0.0075, +0.0076]* | 0.6422 *[-0.0042, +0.0042]* | 32.3355 *[-0.1300, +0.1303]* | 0.3059 *[-0.0027, +0.0028]* | 1.274 |
| LoHA | 0.4244 *[-0.0073, +0.0072]* | 0.6232 *[-0.0041, +0.0041]* | 32.1687 *[-0.1364, +0.1349]* | 0.3009 *[-0.0027, +0.0028]* | 1.268 |
| FourierFT | 0.2153 *[-0.0066, +0.0065]* | 0.5184 *[-0.0041, +0.0042]* | 32.3188 *[-0.1388, +0.1402]* | 0.2495 *[-0.0026, +0.0027]* | 1.173 |
| LoKR | 0.4493 *[-0.0079, +0.0076]* | 0.6323 *[-0.0042, +0.0044]* | 32.5345 *[-0.1336, +0.1333]* | 0.3119 *[-0.0029, +0.0029]* | 1.312 |

Table 10: Sample Variances of Metrics Across Seeds (0-9) for Configuration Groups for the dog instance for LoRA $r = 1$ budget.

| Configuration Group | Var (DINO Sim) | Var (CLIP-I Sim) | Var (CLIP-T Score) | Var (LPIPS Diversity) | Var (CMMD Value) |
|---|---|---|---|---|---|
| SHiRA | 0.00105251 | 0.00009462 | 0.05590719 | 0.99972289 | 0.01821341 |
| WaveFT | 0.00080235 | 0.00008035 | 0.03283789 | 0.99991386 | 0.03002796 |

Table 11: Evaluation Summary for Different $\lambda$ values for LoRA rank=4 equivalent paramter budget with 95% CIs. Confidence intervals are shown below the mean value as [*-lower difference, +upper difference*].

| Configuration Name | DINO Sim ↑ | CLIP-I Sim ↑ | CLIP-T Score ↑ | LPIPS Diversity ↑ | CMMD Value ↓ |
|---|---|---|---|---|---|
| WaveFT $\lambda$=5 | 0.4738 *[-0.0079, +0.0078]* | 0.6430 *[-0.0042, +0.0043]* | 32.5657 *[-0.1334, +0.1316]* | 0.3350 *[-0.0029, +0.0030]* | 1.330 |
| WaveFT $\lambda$=10 | 0.5471 *[-0.0080, +0.0078]* | 0.6803 *[-0.0042, +0.0042]* | 31.9829 *[-0.1320, +0.1343]* | 0.3641 *[-0.0031, +0.0031]* | 1.327 |
| WaveFT $\lambda$=15 | 0.5884 *[-0.0075, +0.0075]* | 0.7024 *[-0.0039, +0.0039]* | 31.5831 *[-0.1357, +0.1389]* | 0.3853 *[-0.0032, +0.0031]* | 1.293 |
| WaveFT $\lambda$=20 | 0.6211 *[-0.0070, +0.0070]* | 0.7196 *[-0.0036, +0.0037]* | 31.0686 *[-0.1394, +0.1377]* | 0.3905 *[-0.0031, +0.0029]* | 1.299 |
| WaveFT $\lambda$=25 | 0.6267 *[-0.0065, +0.0066]* | 0.7131 *[-0.0035, +0.0035]* | 30.3889 *[-0.1358, +0.1359]* | 0.3865 *[-0.0027, +0.0025]* | 1.417 |

### A.3.5 THE EFFECT OF λ ON ADAPTER PERFORMANCE

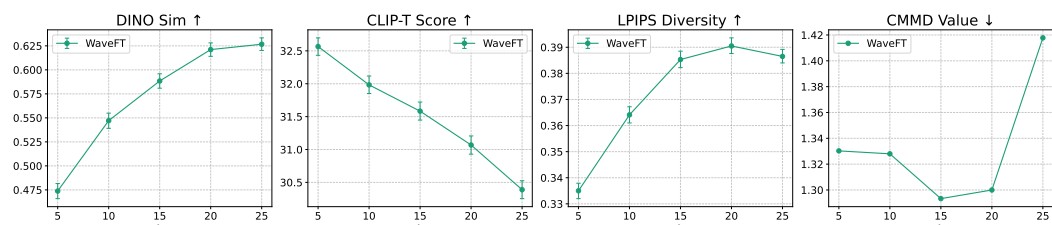

Figure 6: Effect of λ on WaveFT performance (rank-4 equivalent parameters). Increasing λ tends to enhance subject fidelity (DINO, CLIP-I) at the cost of prompt alignment (CLIP-T). This is also consistent with the behavior of other adapter types.

### A.3.6 THE ENERGY DISTRIBUTION OF WAVELET COEFFICIENTS

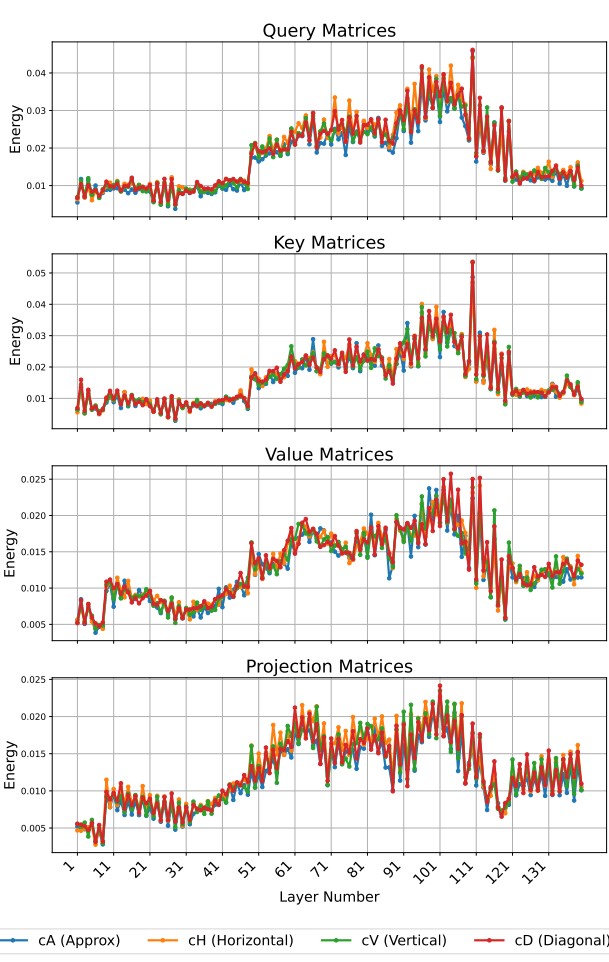

Figure 7: The energy distribution of wavelet coefficients throughout layers. There is no significant bias towards a specific subband.

