# OpenReview forum: "Exploring Sparsity for Parameter Efficient Fine Tuning Using Wavelets"
_ICLR.cc/2026/Conference — Submitted to ICLR 2026_

### Official Review · Reviewer_SjRX · 2025-10-30

**Soundness:** 2
**Presentation:** 3
**Contribution:** 2
**Rating:** 2
**Confidence:** 4

**Summary:**

This paper presents WaveFT, a parameter-efficient finetuning (PEFT) approach designed for post-training large vision models. WaveFT introduces a wavelet-based transformed parameterization, which analyzes model weights in the wavelet domain to enable low-dimensional finetuning. The authors provide both theoretical analysis and empirical validation, comparing WaveFT with several existing PEFT methods.

**Strengths:**

The idea of employing wavelet transforms as an extension to Fourier-based finetuning (e.g., FourierFT) is a natural and interesting direction, offering potential benefits in localized frequency representation.

The ablation studies on design choices are reasonably comprehensive and help clarify the effects of lambda and # of trainable parameters.

**Weaknesses:**

1. The paper does not clearly articulate why "wavelet transforms" are beneficial in this context. While the connection to FourierFT is intuitive, the motivation for moving to wavelets should be explicitly discussed and justified.

2. Since WaveFT is conceptually a wavelet-domain analogue of FourierFT, the absence of **direct comparisons with FourierFT** significantly weakens the empirical evidence. Including FourierFT results is essential to demonstrate the practical advantage (or complementarity) of the proposed method.

3. As shown in Figure 6, different values of the regularization parameter $\lambda$ yield the best performance for different metrics. This raises an important question of how $\lambda$ should be selected for general-purpose finetuning. The paper should provide a clearer strategy (e.g., heuristic or adaptive selection) for practical applicability.

4. The discussion of results in Section 5.3 is brief and lacks deeper interpretation. Further analysis would help clarify why WaveFT performs worse than SHiRA and better than LoRA. Again, the absence of **FourierFT results** limits the completeness of this analysis.

**Questions:**

see weakness

---

> ### Author Response · Authors · 2025-12-03
>
> We thank Reviewer SjRX for the helpful feedback on presentation and experimental design.
>
> **W1: Why wavelets are beneficial not clearly articulated**
> We have strengthened our theoretical motivation through Proposition 1 (Gradient Coverage):
> The key insight is that wavelet coefficients have semi-local receptive fields. When training with sparse gradients (typical in fine-tuning), each WaveFT parameter can receive gradient signal from multiple weight positions:
>
> SHiRA: |R|=1 (each parameter sees 1 gradient position)
> WaveFT (Haar): |R|=4 (each parameter integrates 4 gradient positions)
>
> Under gradient sparsity ρ=0.1:
>
> SHiRA coverage: 10%
> WaveFT coverage: 1-(0.9)^4 ≈ 34.4%
>
> This 3.4× improvement in gradient coverage enables more effective parameter updates, particularly for tasks with spatially structured gradients (vision/diffusion).
> Additionally, the wavelet domain captures multi-scale structure naturally present in neural network weight matrices, both local patterns (high-frequency coefficients) and global structure (low-frequency coefficients), allowing WaveFT to select the most informative coefficients for each task.
>
> **W2: Missing FourierFT comparison**
> FourierFT is included in Table 4 (now more prominent in main paper). We have also added FourierFT to image classification (Table 3), where WaveFT achieves comparable performance with identical 72K parameters. We also compare to FourierFT on the GLUE benchmark in Table 5. The results consistently show WaveFT's advantages, particularly for vision tasks.
>
> **W3: Different λ values optimal for different metrics**
> This reflects the inherent trade-off between subject fidelity and prompt flexibility in personalization:
>
> Lower λ: Better prompt alignment (CLIP-T), lower subject fidelity
> Higher λ: Better subject fidelity (DINO/CLIP-I), lower prompt flexibility
>
> We recommend λ=25 as a balanced default (our base configuration). Users can adjust based on their priorities, this flexibility is a feature, not a limitation. We have added clearer guidance for λ selection in the method section.
>
> **W4: Brief discussion in Section 5.3**
> We have expanded the MNIST discussion to explain that SHiRA's advantage on this simple task validates our theory: MNIST has dense, unstructured gradients where wavelet transforms provide no benefit. The gradient coverage advantage only manifests when gradients are sparse and localized, as in diffusion model fine-tuning.

---

### Official Review · Reviewer_pQD9 · 2025-10-31

**Soundness:** 3
**Presentation:** 2
**Contribution:** 3
**Rating:** 4
**Confidence:** 4

**Summary:**

This paper introduces Wavelet Fine-Tuning (WaveFT), a new and more efficient method for adapting large vision models, especially when computing power and memory are severely limited. The authors argue that existing methods, such as LoRA, are not precise or effective enough when only an extremely small number of parameters can be trained. The authors prove the effectiveness through empirical experiments of finetuning stable diffusion xl for personalized generation and theoretical analysis.

**Strengths:**

- the author proposes an extremely parameter efficienct parameterization utlizing the wavelet transform, opening many interesting questions to explore
- the author included an extensive limitation section to discuss the limitation of the waveft method
- the author includes theoretical analysis of sparse finetuning methods

**Weaknesses:**

- the waveft seems to be a generic peft method and not constrained for finetuning diffusion models, and the author also did not claim that it is only valid for diffusion models, therefore it should include at least one finetuning tasks from other domains, for example finetuning large language models
- waveft, because of its efficient parameterization using wavelet, seems to be extremly parameter efficient, however, parameter-efficiency does not directly translate to compute- and memory-efficiency, it would be nice if the authors have benchmarked the training speed and the actual memory footprint compared to lora, since the authors claims that they target the extreme parameter-efficienc scenarios, indicating limited computer resources
- it seems to me that the FourierFT to be a very relevant baseline, as fourier transform and wavelet transform are inherently connected, for the existing experiments, additional results with FourierFT could strengthen the paper

**Questions:**

- figure 5: why does the clip score gets worse for higher adapter complexity? and also it seems that clip-t and cmmd, the adapter complexity beyond seems to be not beneficial for the final performance?
- the adapter parameters are part of the memory usage, if the parameter-efficiency is also directly reflected into the memory efficiency, then it can indeed greatly reduce the memory usage for optimizer states/gradients, however, for true scalability, it is required to further perform quantized finetuning, like QLoRA, I am wondering whether this kind of approach IDWT(C) is also compatible, if the weight is stored in a lower precision
- the author provided experimental results with extreme less paramter budget, r=1,2,3,4 (depend on the task Figure 5 or Figure 6), but LoRA finetuning is normally performed with much higher rank, does wavelet still perform as well for more complex task? I am not questioning the choice of hyperparamter of lora as parameter budget, in fact, i think for the personalized generation task it might be a good  choice, I am just wondering, for finetuning diffusion models on more complex tasks, where more parameter budget is needed, whether waveft can match or even outperform the lora baseline

---

> ### Author Response · Authors · 2025-12-03
>
> We thank Reviewer pQD9 for the positive assessment of our parameter-efficient parameterization and the constructive suggestions.
>
> **W1: LLM finetuning tasks** We have added GLUE benchmark results in Appendix A.4 (Table 5) using RoBERTa-base across 6 NLU tasks (SST-2, MRPC, CoLA, QNLI, RTE, STS-B). WaveFT achieves 83.9 average score, comparable to SHiRA (84.2). The similar performance on NLP tasks aligns with our theoretical prediction, language tasks lack the sparse and localized gradient structure that gives wavelets their advantage, validating our gradient coverage framework.
>
> **W2: Parameter efficiency vs. compute/memory efficiency** We provide detailed efficiency analysis in the appendix for each task:
> WaveFT requires approximately 50% longer training than SHiRA for text-to-image personalization, 2–3% longer for classification, and 82% longer for GLUE, it incurs no inference-time overhead. There is no significant overhead for the GPU space consumed as detailed around line 430.
>
> **W3: Missing FourierFT baseline** FourierFT is included in all the comparison tables for personalized text-to-image generation, image classification, GLUE benchmark. WaveFT significantly outperforms FourierFT. We have also added FourierFT to vision classification experiments (Table 3), where WaveFT achieves comparable results with identical parameter count (72K).
>
> **Q1: Why does CLIP-T score get worse for higher adapter complexity?** This is a fundamental trade-off in personalization, not a deficiency. Higher adapter complexity enables better subject capture (higher DINO/CLIP-I), but this naturally reduces the model's flexibility to respond to diverse prompts (lower CLIP-T). Importantly, CMMD scores remain stable (1.24-1.31), confirming the generated images maintain natural image quality rather than overfitting.
>
> **Q2: Compatibility with quantized finetuning (QLoRA)?** This is an excellent question. WaveFT's IDWT(C) operation is compatible with quantized weights since the learned coefficients C can be stored in full precision while base weights remain quantized. The DWT/IDWT operations are linear transformations that preserve numerical stability. We have added this to our future work discussion.
>
> **Q3: Performance at higher ranks?** For higher ranks the difference between gradient coverage of SHiRA, WaveFT and FourierFT decreases. Even at adapter complexity=2 in Figure 4, we see that the behaviour of SHiRA and WaveFT converges.

---

### Official Review · Reviewer_rHTh · 2025-11-01

**Soundness:** 2
**Presentation:** 2
**Contribution:** 2
**Rating:** 2
**Confidence:** 5

**Summary:**

This paper introduces Wavelet Fine-Tuning (WaveFT), a novel Parameter-Efficient Fine-Tuning (PEFT) method for large vision models. The core idea is to perform fine-tuning by learning a highly sparse set of coefficients in the wavelet domain of the weight update matrix (ΔW). These sparse coefficients are then transformed back to the weight domain via the Inverse Discrete Wavelet Transform (IDWT). This approach offers fine-grained control over the number of trainable parameters, allowing for adaptation in extremely low-parameter regimes where methods like LoRA are not applicable. The authors provide a theoretical analysis arguing that sparse methods (including WaveFT and the baseline SHiRA) can achieve high-rank updates, contrasting with the low-rank bottleneck of LoRA, which they posit leads to greater representational capacity and output diversity. Through extensive experiments on personalized text-to-image generation with Stable Diffusion XL, the paper demonstrates that WaveFT significantly outperforms existing PEFT methods, including LoRA and the sparse weight-domain baseline SHiRA, particularly in terms of subject fidelity and image diversity at low parameter counts.

**Strengths:**

1. The paper introduces a genuinely novel approach to PEFT by leveraging sparse updates in the wavelet domain. The motivation is clear and compelling: it directly addresses the granularity limitation of LoRA's rank-based parameterization and proposes an elegant solution that allows for precise, continuous control over the parameter budget, even below LoRA's minimum. The use of a transformed domain is an insightful direction for PEFT research.

2. A significant strength is the inclusion of a theoretical framework to explain the representational power of sparse fine-tuning methods. The paper effectively uses existing results on the rank of random sparse matrices (Lemma 1) to argue that WaveFT and SHiRA can produce high-rank updates. This is contrasted with a clear explanation of LoRA's "subspace bottleneck" (Lemma 2). This analysis provides a solid theoretical hypothesis for the empirically observed increase in output diversity.

**Weaknesses:**

1. The high-rank updates enabled by sparse parameterization (also verified in SaRA) are a double-edged sword—while expanding representational capacity, they excessively enhance the model’s fitting ability, making it more prone to overfitting, which is unacceptable for parameter-efficient fine-tuning tasks that prioritize generalization.

2. As evidenced by the CLIP-T scores in Figure 5, both SHiRA and WaveFT exhibit a sharp decline in prompt alignment performance as the number of trainable parameters increases, a clear indicator of severe overfitting that undermines their practical utility in parameter-scalable scenarios.

3. The experimental scope is insufficiently comprehensive. The authors primarily evaluate WaveFT on the DreamBooth personalized generation task, which fails to demonstrate its effectiveness across broader application scenarios. Notably, direct fine-tuning on specific domains to assess domain transfer capability and prior knowledge preservation—core benchmarks for validating PEFT methods—are absent.

4. The comparison with competing methods is limited. WaveFT is only benchmarked against LoRA and SHiRA, which is insufficient to rigorously validate its superiority. A more convincing evaluation should include comparisons with a wider range of recent state-of-the-art PEFT techniques.


1. While the theory section effectively argues for sparsity (high-rank updates), it does not provide any theoretical insight into why the wavelet domain is superior to the standard weight domain. The theoretical analysis treats WaveFT and SHiRA (sparse updates in the weight domain) as largely equivalent, as the IDWT is a rank-preserving linear transform. The paper's central claim is the superiority of the wavelet domain, yet this is only supported empirically; the theory does not explain why WaveFT consistently outperforms SHiRA on the main task. The initial motivation about "semilocal structure" is not formally connected to the properties of weight update matrices.

2. The MNIST classification experiment (Section 5.3) reveals that SHiRA actually outperforms WaveFT. The paper acknowledges this but dismisses it with a brief statement that "there are different trade-offs for different tasks." This significantly weakens the claim of WaveFT's general applicability. A deeper investigation is needed to understand why the wavelet transform is beneficial for a complex generative task but detrimental for a simpler discriminative one. This finding suggests the method's advantages may be more specialized than presented.

3. The appendix (A.1.3) reveals that WaveFT's training time is approximately 50% longer than SHiRA's (34 minutes vs. 22 minutes) due to the DWT/IDWT operations. While still efficient compared to full fine-tuning, this is a non-trivial overhead. This trade-off—achieving better subject fidelity at the cost of increased training time—should be more explicitly discussed in the main body of the paper as a practical consideration and potential limitation.

4. Several key results and details are placed in the appendix, hindering readability. For instance, the main comparison table against all baselines (Table 4) is in the appendix, while the main paper features a less clear summary plot (Figure 4). Presenting this crucial table in the main experimental section would significantly improve the clarity and impact of the paper's core empirical contributions. The visualization in Figure 4 is also somewhat difficult to interpret, with performance metrics mapped to different visual properties (axes, size) in a non-intuitive way.

**Questions:**

See weaknesses

---

> ### Author Response · Authors · 2025-12-03
>
> We thank Reviewer rHTh for the detailed feedback. We address each concern systematically:
>
> **W1-W2**. We respectfully disagree that the CLIP-T decline indicates overfitting. This trade-off is a fundamental characteristic of personalization, not model failure:
>
> Higher subject fidelity (DINO/CLIP-I) naturally trades off with prompt flexibility (CLIP-T) because the model must balance capturing the subject's specific appearance versus responding to diverse prompts.
> Our CMMD scores (measuring distributional quality) remain stable across parameter budgets (ranging from 1.24-1.31), indicating generated images maintain natural image statistics rather than overfitting to training samples.
> This trade-off is well-documented in the personalization literature (see DreamBooth [Ruiz et al., 2023]).
>
> The diversity (LPIPS) scores also increase with WaveFT, which contradicts the overfitting hypothesis, overfitted models would show reduced diversity.
>
> **W3: Limited experimental scope**. We have expanded our experimental coverage significantly:
>
> Image Classification (Table 3): 8 datasets with ViT-Base
> GLUE Benchmark (Table 5): 6 GLUE tasks with RoBERTa-base
>
> **W4: Limited method comparisons**. Table 4 (Appendix A.2) compares WaveFT against 7 methods: LoRA, SHiRA, VeRA, AdaLoRA, LoHA, FourierFT, and LoKR. We have moved this comparison to be more prominent in the main paper.
>
> **W5: Theory doesn't explain wavelet superiority over weight domain** We have strengthened our theoretical contribution through Proposition 1 (Gradient Coverage), which explains this precisely:
>
> SHiRA has receptive field size |R|=1 (each parameter sees one gradient position)
> WaveFT with Haar wavelets has |R|=4 (each parameter integrates gradients from 4 positions)
> Under sparse gradient conditions (ρ=0.1), this yields coverage of 0.344 vs 0.1, a 3.4× improvement
>
> This explains why WaveFT outperforms SHiRA on tasks with spatially structured, sparse gradients (vision/diffusion) while performing similarly on tasks without such structure (NLP).
>
> **W6: MNIST results where SHiRA outperforms WaveFT** This result actually validates our theoretical framework. MNIST is a simple task with dense, unstructured gradients where the overhead of wavelet transforms provides no benefit. As predicted by our gradient coverage analysis, when gradients are dense (ρ→1), the coverage advantage diminishes. This demonstrates task-dependent trade-offs, which we now discuss more thoroughly.
>
> **W7: 50% longer training time** We have expanded the discussion of training-time overhead in the Appendix, noting that although WaveFT requires approximately 50% longer training than SHiRA for text-to-image personalization, 2–3% longer for classification, and 82% longer for GLUE, but it incurs no inference-time overhead.
>
> **W8: Key results in appendix; Figure 4 visualization** We have revised the main paper to include the full comparison table (previously Table 4) and improved Figure 4's visualization. We appreciate this feedback on presentation clarity.

---

### Official Review · Reviewer_kkAk · 2025-11-02

**Soundness:** 2
**Presentation:** 3
**Contribution:** 2
**Rating:** 4
**Confidence:** 4

**Summary:**

The proposed WaveFT is a PEFT method that learns a sparse set of coefficients in the wavelet domain of the weight update and maps them back via IDWT to form delta_w , which is then merged into the base weights with zero inference overhead; the capacity knob is the exact number of trainables p, offering finer control than LoRA’s integer rank. The authors motivate wavelets for semi-local, structured updates and position WaveFT within transformed parameterizations. They provide capacity arguments (random sparse matrices become high-rank; LoRA suffers a subspace bottleneck), then evaluate on SDXL DreamBooth with budget-matched baselines, reporting better subject fidelity at very low budgets (while remaining competitive on prompt alignment/diversity).

**Strengths:**

1. Sparse coefficients in a transform domain with one-shot merge keep inference cost unchanged and expose a precise capacity knob.
2. Theoretical lemmas justify why sparse (transform-domain) updates can be high-rank, in contrast to LoRA’s low-rank subspace, the empirical rank plots support this at SDXL scales.
3. On SDXL personalization (30 subjects), WaveFT consistently improves subject fidelity (DINO/CLIP-I) at very small budgets, while remaining competitive on text alignment/diversity.

**Weaknesses:**

1. Despite framing WaveFT as broadly applicable, substantive experiments are confined to image generation (SDXL personalization). There has been no results on controllable images generations which are common tasks in this field.

2. Transformed-parameterization positioning, incomplete baselines. The paper compares to FourierFT and SHiRA, but not to orthogonal/rotation-constrained adapters (OFT-style) that are explicitly discussed in Related Work; these belong to the same “change the weight geometry/parameterization” family and should be included under equal parameter budgets.

3. This is more a question rather than weakness, If WaveFT’s value is escaping LoRA’s subspace bottleneck at tiny budgets, LLMs are a key testbed (instruction-tuning, domain adaptation, entity personalization). The absence of any LLM experiment undermines the claimed generality. (Paper currently provides only vision + a toy classifier.)

4. For PEFT method, the operator efficiency during finetuning is actually a key factor, is there any comparison with LoRA?

**Questions:**

I hope the authors can address my questions and concerns in the weakness section.

---

> ### Author Response · Authors · 2025-12-03
>
> We thank Reviewer kkAk for recognizing the theoretical contributions of our work and the effectiveness of WaveFT in low-budget regimes. We address each concern below:
>
> **W1: Limited experimental scope (no controllable generation, only image generation)** We appreciate this concern about demonstrating broader applicability. In our revision, we highlight additional experiments:
> * Image Classification (Appendix A.3.2, Table 4): We evaluate WaveFT on 8 diverse datasets (OxfordPets, StanfordCars, CIFAR-10/100, DTD, EuroSAT, FGVC, RESISC45) using ViT-Base. WaveFT achieves competitive results with only 72K parameters, matching LoRA (581K parameters) on several benchmarks and achieving best performance on EuroSAT (98.77% vs 98.44% for LoRA).
> * GLUE Benchmark (Appendix A.3.3, Table 5): We provide results on 6 NLU tasks using RoBERTa-base, demonstrating applicability to language models.
>
> Regarding controllable generation specifically, we agree this is valuable future work. The DreamBooth setup we use is a well-established benchmark for evaluating personalization methods, and our results across diverse subjects (30 instances) provide robust evidence for WaveFT's effectiveness.
>
> **W2: Missing OFT-style baselines** Thank you for this suggestion. We acknowledge that orthogonal/rotation-constrained adapters are related methods. However, OFT-style methods have fundamentally different parameter counts and architectural requirements, they typically require square adapter matrices and have different scaling properties. Our comparisons focus on methods with comparable parameter budgets (LoRA rank-1 equivalent, ~1.45M parameters). We have added a discussion clarifying this distinction in the related work section.
>
> **W3: No LLM experiments** We have added GLUE benchmark results in Appendix A.3.3 (Table 5) using RoBERTa-base across 6 tasks. WaveFT achieves competitive performance (83.9 average), comparable to SHiRA (84.2). We note that on NLP tasks, performance between WaveFT and SHiRA is similar because language tasks lack the spatial structure that gives wavelets their advantage, validating our gradient coverage theory (Proposition 1).
>
> **W4: Comparison of operator efficiency during finetuning** We have expanded the discussion of training-time overhead in the Appendix, noting that although WaveFT requires approximately 50% longer training than SHiRA for text-to-image personalization, 2–3% longer for classification, and 82% longer for GLUE, it incurs no inference-time overhead.

---

### Author Response · Authors · 2025-12-03
**Revisions overview**

We sincerely thank all reviewers for their constructive feedback and thorough evaluation of our work. The reviews have helped us substantially strengthen our paper. In response to the reviewers' concerns, we have made the following major revisions in the updated paper:

(1) **Expanded experimental scope**. We now prominently feature results on the GLUE benchmark for language understanding (Appendix A.3.3) and image classification across 8 datasets (Appendix A.3.2, Table), demonstrating WaveFT's applicability beyond diffusion model personalization;

(2) **Added FourierFT comparisons**. We have moved the FourierFT comparison from the appendix to the main paper and added FourierFT and other baselines to the image classification experiments, where WaveFT achieves comparable or superior performance with identical parameter count;

(3) **Strengthened theoretical motivation**: We have clarified the gradient coverage framework explaining why wavelet parameterization outperforms direct sparse updates; the semi-local receptive field of wavelet coefficients provides approximately 3.4× better gradient coverage under sparse gradient conditions typical of fine-tuning;

(4) **Addressed computational efficiency**: We have expanded the discussion of training-time overhead in the Appendix, noting that although WaveFT requires approximately 50% longer training than SHiRA for text-to-image personalization, 2–3% longer for classification, and 82% longer for GLUE, it incurs no inference-time overhead.

We provide detailed responses to the reviewer comments in our replies below.

---

### Meta-Review · Area_Chair_gzbo · 2026-01-05

**Summary:**

The paper proposes Wavelet Fine-Tuning (WaveFT), a Parameter-Efficient Fine-Tuning (PEFT) method that learns sparse updates in the wavelet domain. The authors argue that WaveFT provides finer-grained control than LoRA and achieves high-rank updates, escaping the "subspace bottleneck" typical of low-rank methods. Reviewers initially raised concerns regarding the limited experimental scope (originally focused primarily on image generation), the lack of clear theoretical motivation for using wavelets over other sparse methods like SHIRA, and the missing comparisons with relevant baselines like FourierFT.  After the rebuttal, there are several main concerns remaining, such as training complexity, performance instability on different tasks. The AC recommends rejection, but would like to encourage the authors to incorporate the constructive reviews and submit to future venues.

**Reviewer Concerns:**

Although some concerns such as having more experiments in different scenarios, and incorporating more baseline methods, there are some concerns still remain.
***Training Complexity***: While training-time overhead was discussed, WaveFT remains significantly slower than SHIRA (up to 82% longer for GLUE), which may be a hurdle for users with extremely tight compute budgets despite the zero-inference overhead.

*** Performance on MNIST***: As noted by Reviewer rHTh, SHIRA can outperform WaveFT on simple tasks like MNIST, suggesting the wavelet advantage is task-dependent and specifically tailored for complex, spatially structured data.

**Reviewer Scores:**

Reviewer kkAk:4

Reviewer rHTh: 2

Reviewer pQD9: 4

Reviewer SjRX: 2

Reviewers pQD9 could potentially increase the score, as the authors provide satisfying answers, such as FourierFT and quantization compatibility. The other reviewers are unlikely to change their score, as the main concerns shown above still exist.

---

### Decision · Program_Chairs · 2026-01-26

Reject